# The cotranslational cycle of the ribosome-bound Hsp70 homolog Ssb

Ying Zhang [1,7], Lorenz Grundmann [2,3,7], Leonie Vollmar[4,5], Julia Schimpf[4,5], Volker Hübscher [1], Mohd Areeb[1,5], Irina Grishkovskaya [2], Anna Moddemann[1], Kerstin Werner[1], Thorsten Hugel [4,6], David Haselbach [2] ✉ & Sabine Rospert [1,6] ✉

Coupling of ribosomal translation with cotranslational protein folding is essential for cellular homeostasis. In eukaryotes, Hsp70 and its J-domain cochaperone, the heterodimeric ribosome-associated complex (RAC), are central to this process; however, mechanistic insights into the coordination of Hsp70 function with translation remain limited. Here, we present two cryo-EM structures of the ribosome-bound yeast Hsp70 Ssb, identifying Rpl25/uL23 as the ribosomal binding site and revealing its interaction with a model nascent chain. Together with detailed biochemical and mutational analyses, these structures enable us to delineate the intricate RAC-dependent cycle, which positions the substrate binding domain of Ssb-ATP close to the tunnel exit to receive nascent chains. This arrangement allows Ssb to undergo substantial conformational changes upon ATP hydrolysis without steric clashes with the ribosome, while the substrate binding domain of Ssb, now anchored by the tightly bound nascent chain, remains close to the tunnel exit.

Cotranslational processes - such as folding, complex assembly, and membrane translocation - critically rely on Hsp70 chaperones. Canonical Hsp70s cooperate with J-domain (JD) cochaperones, which stimulate their ATPase activity and substrate binding, and with nucleotide exchange factors (NEFs), which facilitate ADP-release and rebinding of ATP[1–4] (Supplementary Fig. 1a). Yeast Ssb is a canonical Hsp70 (encoded by the nearly identical *SSB1* and *SSB2* genes), consisting of an N-terminal nucleotide binding domain (NBD) and a C-terminal substrate binding domain (SBD) (Fig. 1a). The NBD possesses ATPase activity, the SBD is subdivided into a β-sheet domain (SBDβ) and an α-helical lid domain (SBDα) (Fig. 1a). As in other canonical Hsp70s, the two domains of Ssb are connected via a linker that allows allosteric coupling of ATP hydrolysis with tight substrate binding to the SBD (Supplementary Fig. 1a). When Ssb is bound to ATP,

the linker interacts with the NBD, and the SBDα lid adopts an open conformation[5]. Upon ATP hydrolysis, significant structural rearrangements occur, which lead to the detachment of the linker from the NBD, allowing the SBDα lid to close over the substrate binding pocket on SBDβ, thereby stabilizing substrate binding[1,4,6].

Ssb is critically important for cotranslational protein folding, as it binds directly to ribosomes via SBD[5,7] and, from this position, accesses newly synthesized polypeptides[8–10]. To perform its function, Ssb requires a unique cochaperone, termed ribosome-associated complex (RAC)[6,11,12]. The heterodimeric RAC consists of the JD protein Zuo1[13] and the Hsp70 homolog Ssz1[6,11]. The Zuo1 subunit interacts with the 60S as well as 40S ribosomal subunit[6,14–17] and possesses a unique domain structure[6,16,17] (Supplementary Fig. 1b). Ssz1 is a non-canonical Hsp70 as it binds, but does not hydrolyze ATP, and lacks the

[1]Institute of Biochemistry and Molecular Biology, ZBMZ, Faculty of Medicine, University of Freiburg, Freiburg, Germany. [2]Research Institute of Molecular Pathology (IMP), Campus Vienna Biocenter 1, Vienna, Austria. [3]Vienna BioCenter PhD Program, Doctoral School of the University of Vienna and Medical University of Vienna, Vienna, Austria. [4]Institute of Physical Chemistry, University of Freiburg, Freiburg, Germany. [5]Spemann Graduate School of Biology and Medicine (SGBM), University of Freiburg, Freiburg, Germany. [6]BIOSS Centre for Biological Signalling Studies, and CIBSS Centre for Integrative Biological Signalling Studies, University of Freiburg, Freiburg, Germany. [7]These authors contributed equally: Ying Zhang, Lorenz Grundmann. ✉e-mail: david.haselbach@imp.ac.at; sabine.rospert@biochemie.uni-freiburg.de

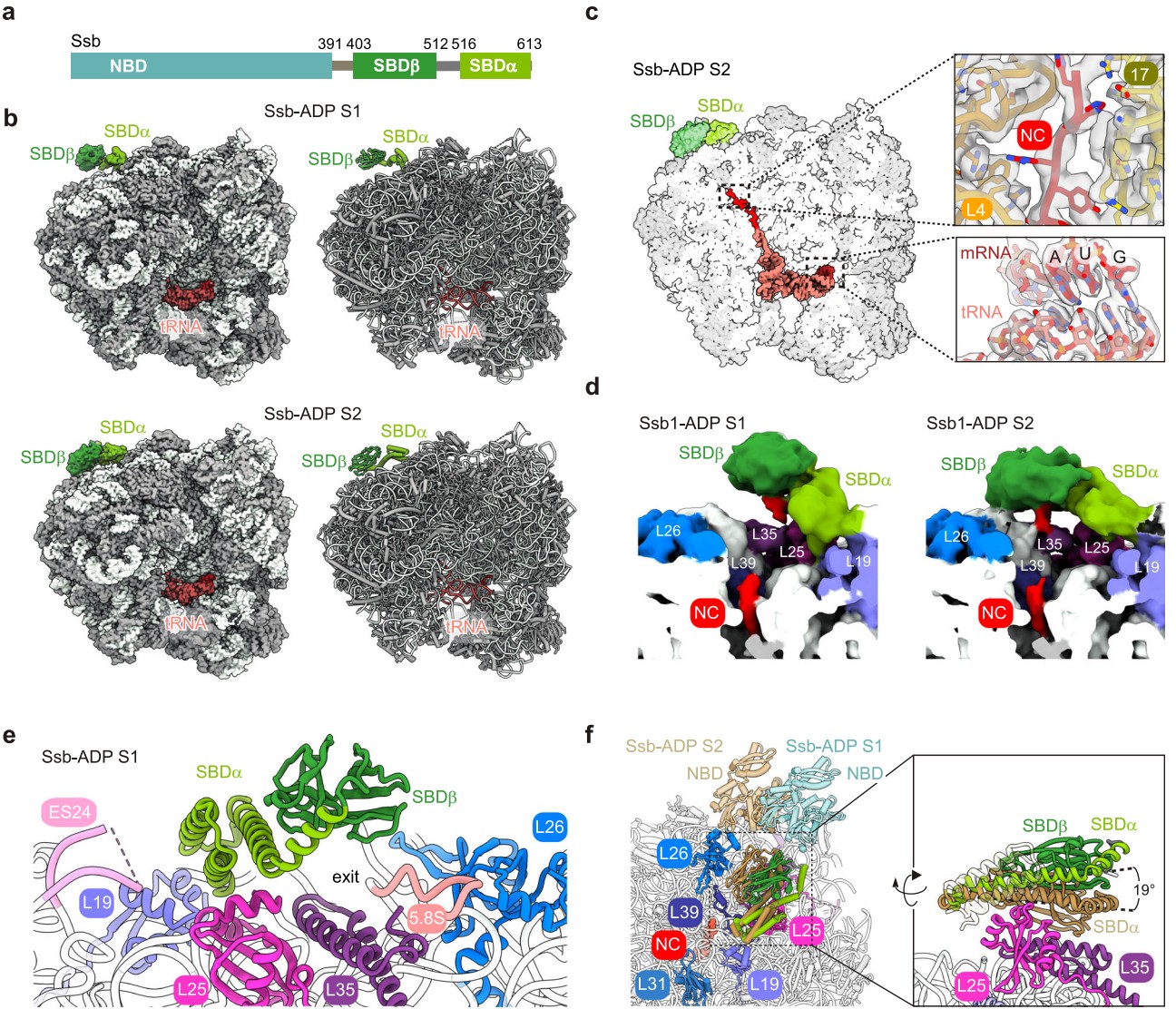

**Fig. 1 | Architecture of Ssb bound to ribosome nascent chain complexes (RNCs).**
**a** Domain structure of Ssb. Nucleotide binding domain (NBD) with ATPase activity, and substrate binding domain (SBD), which is subdivided into a β-sheet domain (SBDβ) and an α-helical lid domain (SBDα). Ssb-NBD (light teal), Ssb-SBDβ (forest) Ssb-SBDα (split pea), linker region (gray). **b** Cryo-EM maps and atomic models of Ssb-ADP S1 and Ssb-ADP S2. RNCs arrested on truncated mRNA in complex with peptidyl tRNA$^{Met-e}$ and ADP-bound Ssb in two conformations termed S1 and S2. **c** Cross section through the cryo-EM map of Ssb-ADP S2. Densities of the nascent chain, mRNA and tRNA$^{Met-e}$ are highlighted. Insets show overlays of cryo-EM densities and atomic models of the nascent chain inside the exit tunnel and tRNA-mRNA base pairing at the peptidyl transferase center. **d** Binding of the nascent chain to the peptide-binding cleft of Ssb-SBDβ. Cross section through lowpass

filtered (sdev=2.5), non-post-processed maps of Ssb-ADP S1 and S2. **e** Close-up of the contacts of Ssb-ADP S1 with the tunnel exit region. A detailed evaluation of contacts is provided in Supplementary Note 1 and Supplementary Fig. 5a, b.
**f** Positioning of the Ssb-SBD relative to the ribosomal tunnel exit region. Top view of Ssb-ADP S1 (Ssb-SBDβ forest, Ssb-SBDα split pea) and Ssb-ADP S2 (sand). The NBDs, not resolved in the cryo-EM maps, were generated by superimposition of Ssb-ADP S1 (transparent light teal) and S2 (transparent sand) with the structure of Ssb-ADP predicted by I-TASSER based on PDB 2KHO (see Methods) for illustrative purposes. The inset shows sides views of Ssb-ADP S1 and S2. Color code of ribosomal proteins and rRNA: Rpl4 (tv orange), Rpl17 (olive), Rpl25 (light magenta), Rpl35 (violetpurple), Rpl39 (deepblue), Rpl26 (marine), Rpl19 (slate blue), Rpl31 (sky blue), helix 59 expansion segment 24 (ES24, pink), 5.8S rRNA (5.8 S, salmon).

C-terminal portion of SBDβ and the complete SBDα[6,18,19] (Supplementary Fig. 1b). In the absence of ribosomes, Ssz1 and Zuo1 engage in a tight interaction, mediated by the N-terminal domain of Zuo1 and the rudimentary Ssz1-SBDβ[6,18–20]. However, when bound to ribosomes, RAC adopts a highly flexible structural arrangement, as evidenced by the markedly different conformations observed in ribosome-bound RAC through cryo-EM studies[16,17]. Conformational flexibility within RAC is highlighted by the extreme N-terminus of Zuo1, termed the LP-motif (Zuo1-LP; Supplementary Fig. 1b[21]), which binds to Ssz1-SBDβ as a pseudo-substrate but is displaced upon nascent-chain binding[21]. Displacement of the Zuo1-LP occurs as soon as the nascent chain emerges from the ribosomal tunnel, thereby loosening the Zuo1-Ssz1

interaction and permitting conformational rearrangements of the N-terminal Zuo1 domain (Zuo1-ND; Supplementary Fig. 1b). Shortly thereafter, the nascent chain is handed over from Ssz1-SBDβ to Ssb-SBDβ, but how this transfer is coordinated remains unknown[21].

RAC possesses a mammalian homolog, termed mRAC, which consists of MPP11/DNJC2[22,23] and HSPA14/Hsp70L1[22,24]. RAC and mRAC are strictly required for cotranslational Hsp70 function[11,12,22,24,25]. However, why Ssb function depends on such a complex JD partner - whereas Hsp70s typically cooperate with structurally simpler monomeric or homodimeric JD proteins[26,27] - remains unresolved. A major obstacle has been the lack of structural information on Hsp70s bound to translating ribosomes and nascent chains[17,28]. In this work, we

provide structural insights into how Ssb interacts with ribosomes and nascent chains, and elucidate the interplay between Ssb and RAC during cotranslational protein folding.

## Results

### Cryo-EM structure of ADP-bound Ssb associated with translating ribosomes

To determine the structure of Ssb associated with translating ribosomes, ribosome-nascent chain complexes (RNCs) carrying FLAG-tagged 3-phosphoglycerate kinase (FLAG-Pgk1-70; Supplementary Fig. 1c–f) were generated in a yeast in vitro translation system in the presence of Ssb, were subsequently purified by FLAG affinity chromatography, and were analyzed by cryo-EM (Supplementary Fig. 2, see Methods). In these samples, ribosomes representing diverse states along the translational cycle were detected. Notably, non-rotated RNCs containing a P-site tRNA and associated with the density of a trailing ribosome displayed enhanced density definition near the ribosomal tunnel exit (Supplementary Fig. S3). Further analysis of these particles (Supplementary Fig. S2) yielded two cryo-EM structures in which Ssb was bound to ribosomes stalled in a non-rotated, post-translocation conformation with a P-site Met-elongator tRNA (tRNA$^{Met-e}$). In both structures, the Ssb-SBD (Fig. 1b and Supplementary Table 1) was in the closed conformation characteristic for the ADP-bound state of Hsp70s, in which SBDα is tightly packed against SBDβ (see Introduction). The Ssb-NBD was not visible in the cryo-EM maps, consistent with the known flexibility between the NBD and SBD in ADP-bound Hsp70s[4,29,30]. The resolution of the complexes, termed Ssb-ADP S1 (2.8 Å) and Ssb-ADP S2 (3.0 Å), allowed precise visualization of tRNA-mRNA interactions, including Watson-Crick base pairing between tRNA$^{Met-e}$ and its cognate codon (Fig. 1b,c, Supplementary Fig. 1e, Supplementary Fig. 4a–h). The density of nascent FLAG-Pgk1-70 was well defined within the proximal region of the exit tunnel, allowing for side chain assignment of a large fraction of the 17 C-terminal residues and enabling confident modeling of the nascent chain (Fig. 1c and Supplementary Fig. 1e). While the nascent chain remained traceable through the constriction site formed by loops of Rpl17 (uL22) and Rpl4 (uL4) in the exit tunnel, its structural definition decreased progressively (Fig. 1c, d, and Supplementary Fig. 1f). Local resolution of the Ssb-SBD enabled side chain assignment for SBDα. In case of SBDβ, flexibility relative to SBDα and the translating ribosome hindered side chain assignment; therefore, the SBDβ backbone was modeled as a rigid body (Supplementary Fig. 4d, h). Ssb-ADP was attached to the ribosome mainly via SBDα with the peptide-binding cleft of SBDβ positioned close to the tunnel exit (Fig. 1e and Supplementary Fig. 5a, b). The inherent flexibility of nascent FLAG-Pgk1-70 bound to Ssb-SBDβ prevented direct visualization at high resolution, however, low-pass filtered non-sharpened maps revealed an extension leading away from the ribosomal tunnel exit to the SBDβ substrate binding cleft, which we tentatively attributed to the nascent chain (Fig. 1d and Supplementary Fig. 1f, red density).

The combined data suggested that both Ssb-ADP S1 and S2 corresponded to ribosome-bound Ssb engaged with a nascent chain substrate (Fig. 1b,d and Supplementary Fig. 1e,f). In both states, Ssb-SBD adopted the closed conformation of an ADP-bound Hsp70 (Supplementary Fig. 1a), but differed in the position of SBDβ relative to the translating ribosome, which was 19° closer to the ribosomal surface in S2 when compared to S1 (Fig. 1f and Supplementary Fig. 5a,b).

Previous studies[5,7] employed crosslinking approaches and mutational analyses to identify contact sites between Ssb and the ribosome. These studies revealed that the most C-terminal helix of Ssb (Ssb-αD, residues 592-611, Supplementary Fig. 5c) is essential for ribosome-binding[5,7]. On the ribosomal side, Rpl35 (uL29), Rpl19 (eL19), Rpl39 (eL39)[5], Rpl26 (uL24)[28], and the 25S rRNA helix 59 expansion segment 24 (h59-ES24)[5] were found to crosslink to Ssb. While in the atomic model of Ssb-ADP S1 the distance of the SBD from Rpl35, Rpl19, Rpl39,

and Rpl26 was inconsistent with crosslinking, the position of the SBD in Ssb-ADP S2 reveals the precise molecular basis of these crosslinks (Fig. 1f, Supplementary Fig. 5a,b, and Supplementary Note 1). Only two ribosomal binding sites were shared by Ssb-ADP S1 and S2. One was the contact of Ssb-K567 and Ssb-R568 with h59-ES24 (Supplementary Fig. 5a,b). This contact provides the molecular basis for the observation that substituting K567-R568 with glutamates partly impairs Ssb´s ribosome-binding[5] (Supplementary Fig. 5c). The other was the contact of Ssb-αD with Rpl25 (Fig. 1e and Supplementary Fig. 5a,b), which had not been identified previously.

### Ssb interacts with ribosomal protein Rpl25 at the ribosomal tunnel exit

Analyses of the Ssb-ADP ribosome interface in the atomic models of states S1 and S2 revealed that Ssb primarily contacts Rpl25 through salt bridges and long-range ionic interactions (Fig. 2a and Supplementary Fig. 6a). Key contributions to the interface came from Ssb residues R596, K597, K603, and R604 (termed the Ssb-RKKR motif, Supplementary Fig. 5c) and Rpl25 residues E77, D131, D134 (termed the Rpl25-EDD motif) and the Rpl25 carboxyl terminus (Fig. 2a and Supplementary Fig. 6a). Specifically, salt bridges were observed between Ssb-R604 and Rpl25-D131, Ssb-R596 and Rpl25-D134, and between Ssb-K603 and the C-terminus of Rpl25 (I142). Long-range ionic interactions were formed between Ssb-K597 and Rpl25-D134, and Ssb-K608 and Rpl25-E77. Although the overall geometry of the binding interface did not differ substantially between S1 and S2, Ssb-SBDβ was closer to the ribosomal surface in S2 than in S1 (Fig. 1f and Supplementary Fig. 5a,b). To validate the interface, we mutated the EDD-motif to three alanine (termed Rpl25-AAA) or three lysine residues (termed Rpl25-KKK) (Supplementary Fig. 6b) and compared ribosome-binding of Ssb to wild type and Rpl25-AAA/KKK ribosomes in total cell extracts. Consistent with earlier findings[5], approximately 50% of wild type Ssb was ribosome-bound, with about two-thirds of this fraction exhibiting salt-resistant binding (Fig. 2b and Supplementary Fig. 6c). Ribosome-binding of Ssb was reduced more than 2-fold in extracts of the Rpl25-AAA and Rpl25-KKK strains under low- and high-salt conditions (Fig. 2b and Supplementary Fig. 6c). To further validate the role of the Rpl25 EDD-motif, we used recombinant Ssb and purified non-translating ribosomes (Supplementary Fig. 6d–f). Wild type Ssb was bound to non-translating ribosomes, however, in this case high-salt conditions led to complete ribosome-release (Fig. 2c,d). The observation suggested that salt-resistant ribosome-binding involved the interaction between Ssb and the nascent chain (see below). Binding of Ssb to purified Rpl25-KKK ribosomes was strongly reduced, while binding of RAC (control) remained unaffected (Fig. 2c). These data established Rpl25 as the primary ribosomal attachment site of Ssb and demonstrated that residues E77/D131/D134 in Rpl25 are crucial for the interaction. In retrospect, the absence of Ssb-Rpl25 crosslink products[5,28] is evident: lysine residues are abundant in both Ssb and Rpl25, however, Rpl25 lacks solvent-exposed lysine residues in close proximity to Ssb-SBD.

The Rpl25-AAA and Rpl25-KKK strains exhibited severe growth defects (Supplementary Fig. 6b), exceeding the only minor growth defects of Ssb mutant strains with significant ribosome-binding defects[5,7]. A potential explanation emerged from Rpl25's established role as a critical binding site for signal recognition particle (SRP), which requires this interaction for efficient cotranslational targeting[31–34]. Structural superposition of Ssb-ADP S2 with SRP-bound RNCs[35] revealed that Ssb and SRP binding would be mutually exclusive (Supplementary Fig. 6g). We tested if targeting of the type II membrane protein Dap2, a well-characterized in vitro and in vivo substrate of SRP[36,37], was affected by mutations in the Rpl25 EDD-motif. Indeed, expression of Dap2 in a Rpl25-KKK strain was significantly reduced when compared to wild type and resembled expression of Dap2 in a Δsrp54 strain (Supplementary Fig. 6h). Moreover, Ssb which was efficiently crosslinked to nascent Pgk1 (Supplementary Fig. 1d and

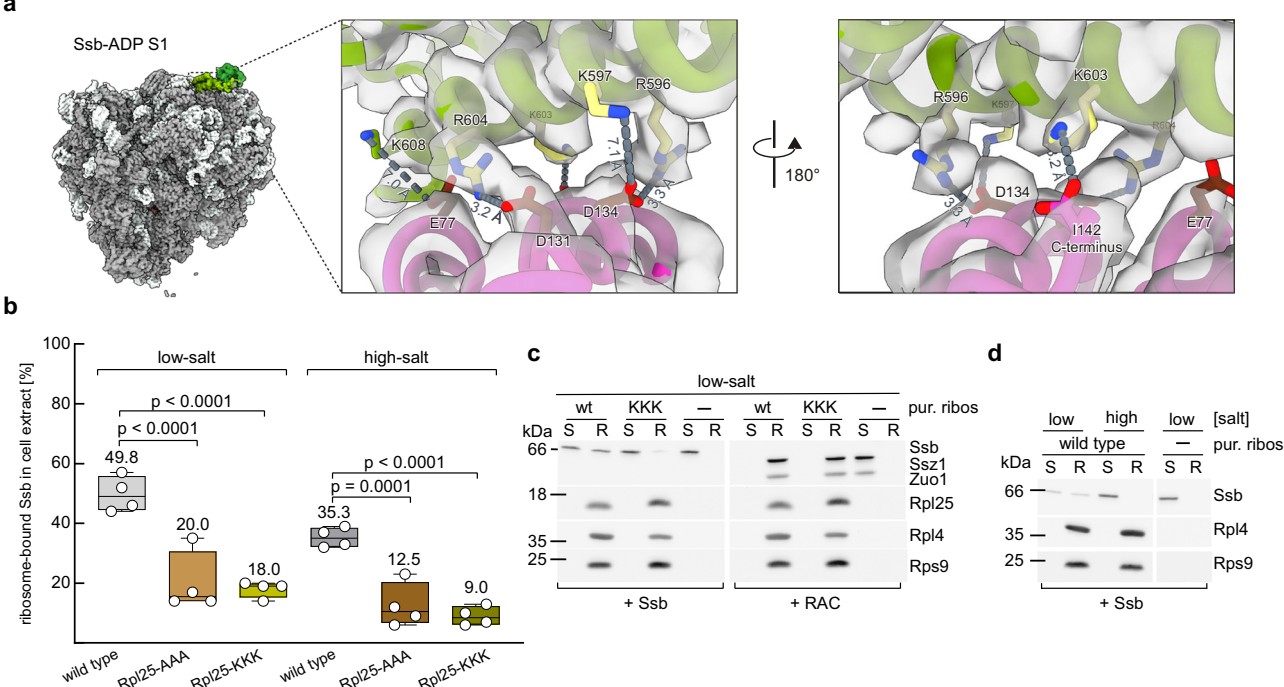

**Fig. 2 | The interaction between Ssb and Rpl25 is essential for efficient ribosome-binding of Ssb. a** The RKKR-motif within helix Ssb-αD contacts the EDD-motif within Rpl25. DeepEMhanced cryo-EM map of Ssb-ADP S1 and close-ups of the contact zone depicting overlays of the cryo-EM map and atomic model. Residues of the RKKR-motif (yellow, Supplementary Fig. 5c), Rpl25 (light magenta) residues of the EDD-motif within Rpl25 (chocolate). Molecular interactions and corresponding distances are depicted in dark gray. **b** Mutations within the Rpl25 EDD-motif significantly reduce ribosome-binding of Ssb. Shown is the fraction of Ssb cosedimenting with ribosomes in total cell extract derived from wild type, Rpl25-AAA, or Rpl25-KKK cells. Two-way ANOVA was performed with data from four independent experiments (dots), as exemplified in Supplementary Fig. 6c. Boxes indicate the 25th to 75th percentiles; lines represent the minimum, median, and maximum values; values shown are means. **c** Ribosome-binding of Ssb to purified Rpl25-KKK ribosomes is strongly reduced. Purified Ssb was incubated with purified, non-translating ribosomes (pur. ribos) derived from wild type or Rpl25-KKK cells. Purified RAC served as a control. Supernatants (S) and ribosomal pellets (R) were analyzed by immunoblotting with antibodies directed against the indicated proteins. The experiment was independently performed twice, yielding similar results. **d** Ribosome-binding of Ssb to purified wild type ribosomes is salt sensitive. Ribosome-binding assays were performed with purified components as in (**c**). The experiment was independently performed twice, yielding similar results.

Supplementary Fig. 6i), did not crosslink to nascent Dap2 under the same conditions (Supplementary Fig. 6i[37]). This pattern changed in the absence of SRP: under this condition Ssb was efficiently crosslinked to nascent Dap2 (Supplementary Fig. 6i). The data support a model in which SRP and Ssb compete for an overlapping ribosomal attachment site involving the Rpl25 EDD-motif. They further suggest that the competition of SRP and Ssb for nascent chains, observed in a global approach[38], is unlikely to result from a general inability of Ssb to interact with SRP substrates, but may rather reflect direct competition at the tunnel exit. Importantly, the impaired delivery of the SRP-dependent ER model protein Dap2 (Supplementary Fig. 6h) provides a plausible explanation for the severe growth defects observed in strains carrying mutations in the Rpl25 EDD-motif (Supplementary Fig. 6b).

### Rpl25 provides the binding site for Ssb in the ATP- and ADP-bound state

To further probe the electrostatic nature of the Ssb-Rpl25 interaction, we engineered a charge-reversal variant of Ssb (Ssb-DD1-DD2, Supplementary Fig. 7a) by replacing the basic residues of the RKKR-motif with aspartates (Fig. 3a). Analysis of Ssb-DD1-DD2 revealed a striking context-dependent binding pattern. In cell extracts containing actively translating ribosomes, Ssb-DD1-DD2 showed significantly reduced but detectable ribosome association (Fig. 3b,c). Under these conditions, Ssb-DD1-DD2 binding was fully salt-resistant, suggesting that the interaction was primarily mediated by the nascent chain (Fig. 3b,c). Consistently, with purified non-translating ribosomes, binding was

completely abolished (Supplementary Fig. 7b and Fig. 2d), highlighting the contribution of nascent chains to ribosome-binding of Ssb.

The combined data revealed that replacing either the positively charged Ssb RKKR-motif with negatively charged residues (Fig. 3b,c) or substituting the negatively charged Rpl25 EDD-motif with positively charged residues (Fig. 2b,c) exerted a similar effect on ribosome-binding of Ssb. We thus performed a charge reversal binding experiment with recombinant Ssb-DD1-DD2 (Supplementary Fig. 6f) and non-translating Rpl25-KKK ribosomes (Supplementary Fig. 6e). Indeed, binding of Rpl25-KKK to Ssb-DD1-DD2 was improved when compared to binding of wild type Ssb confirming the electrostatic nature of the interaction (Fig. 3d). Thus, the Rpl25 EDD- and Ssb RKKR-motifs were not only proximal to each other, as shown in the atomic models Ssb-ADP S1 and S2 (Fig. 2a and Supplementary Fig. 6a), but also essential for Ssb´s direct association with ribosomes.

It has been proposed that Ssb, particularly in its ATP-bound state, does not bind directly to the ribosome but is instead recruited through the dimerization of its NBD with the NBD of Ssz1[2,28]. However, in total cell extract, an ATPase-deficient Ssb mutant was bound to ribosomes, albeit less efficiently compared to wild type Ssb[5]. The RKKR-motif within helix Ssb-αD (Fig. 3a) is prominently exposed when Ssb is bound to ATP[5] (PDB 5TKY). It thus appeared reasonably possible that Ssb-ATP was able to interact with ribosomes via the Rpl25 EDD and Ssb RKKR contact. To investigate this possibility, we superimposed Ssb-αD from the ATP-bound structure (PDB 5TKY[5]) onto Ssb-ADP S1 (Supplementary Fig. 7c). In the resulting Ssb-ATP model, Ssb-SBDβ and Ssb-NBD

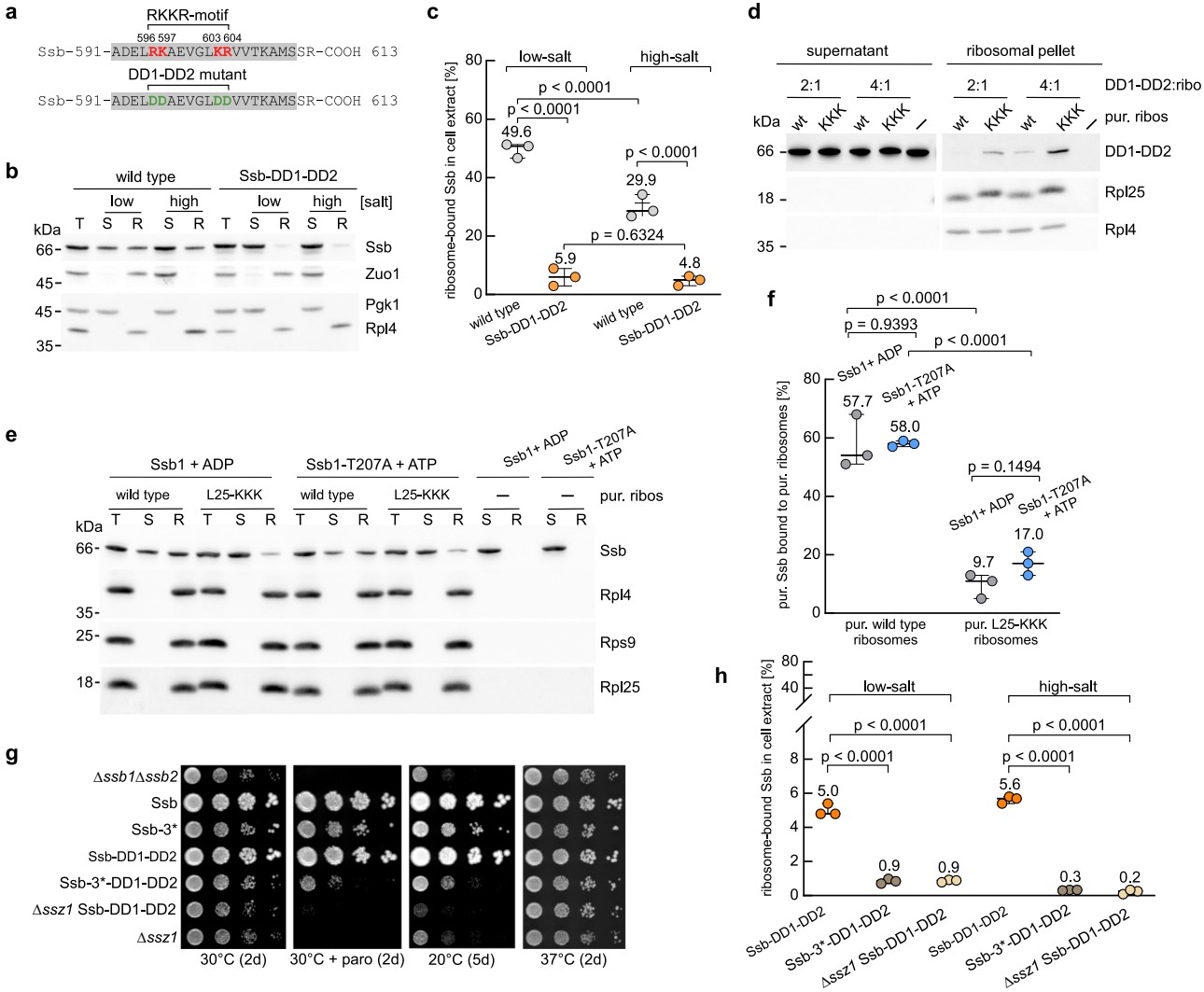

**Fig. 3 | The Ssb RKKR-motif is required for ribosome-binding of Ssb-ADP and Ssb-ATP. a** C-terminal helix Ssb-αD (gray) of wild type Ssb with the RKKR-motif (red) and of the DD1-DD2 mutant (green). **b** Ribosome-binding of Ssb-DD1-DD2 is strongly reduced in total cell extract. Binding assays with extracts from wild type or Ssb-DD1-DD2 cells were performed under low- or high-salt conditions (see Supplementary Fig. 6c). Samples were analyzed by immunoblotting for the indicated proteins; Zuo1 (RAC) served as a ribosome-binding control. **c** Two-way ANOVA was performed with data from three independent experiments (dots), as exemplified in (**b**). Lines represent the minimum, mean, and maximum values; values shown are means. **d** Ribosome-binding of Ssb-DD1-DD2 to Rpl25-KKK ribosomes is enhanced. Purified Ssb-DD1-DD2 (4 or 8 μM) was incubated with wild type or Rpl25-KKK ribosomes (2 μM) and ribosomes were collected by centrifugation under low-salt conditions. A control lacked ribosomes. Supernatants correspond to 50% (2:1) and 25% (4:1) loadings relative to ribosomal pellets. The experiment was independently performed twice, yielding similar results. **e** Ribosome-binding of Ssb-ADP and Ssb-

ATP depends on the Rpl25 EDD-motif. Ribosome-binding assays were performed under low-salt conditions with wild type or Rpl25-KKK ribosomes and purified wild type Ssb (pre-incubated with ADP) or Ssb-T207A (pre-incubated with ATP). Immunoblots were analyzed with antibodies directed against the indicated proteins. **f** Two-way ANOVA was performed with data from three independent experiments (dots), as exemplified in (**e**). Lines represent the minimum, mean, and maximum values; values shown are means. **g** Synthetic growth defects caused by combined mutations in Ssb impairing direct and Ssz1-mediated ribosome-binding. Serial 10-fold dilutions of logarithmically growing cells were spotted onto YPD plates. Plates were incubated for the indicated times and temperatures; paro (50 μg/ml paromomycin). **h** Mutations in Ssb impairing direct and Ssz1-mediated ribosome-binding act synergistically. Two-way ANOVA was performed with data from three independent experiments (dots), as exemplified in Supplementary Fig. 7e. Lines represent the minimum, mean, and maximum values; values shown are means.

are detached from the ribosome and the peptide-binding cleft of Ssb-SBDβ is turned away from the tunnel exit (Supplementary Fig. 7c). To test the interaction of Ssb-ATP with ribosomes directly, we investigated whether purified Ssb-ATP interacts with ribosomes, and if this interaction depended on the Rpl25 EDD-motif. To prevent ATP hydrolysis during the experiment, we employed the Ssb-T207A mutant (Supplementary Fig. 7a), which is incapable of hydrolyzing ATP[5,39,40]. A side-by-side comparison of purified Ssb in the presence of ADP and purified Ssb-T207A in the presence of ATP revealed that the affinity of wild type and mutant for non-translating ribosomes was rather similar (Fig. 3e,f, and see below). Moreover, binding of Ssb-ADP as well as Ssb-

T207A-ATP to purified Rpl25-KKK ribosomes was significantly reduced (Fig. 3e,f). We conclude that Ssb-ATP directly interacts with non-translating ribosomes and that this interaction depends crucially on the RKKR-motif within Ssb and the EDD-motif within Rpl25.

Direct ribosome-binding of Ssb-DD1-DD2 was severely impaired (Fig. 3b,c), yet did not affect yeast growth (Supplementary Fig. 7a). It was previously shown that Ssb-ATP can also associate with ribosomes indirectly, through interaction of the Ssb-NBD with the NBD of the RAC subunit Ssz1[28]. Mutations in Ssb (R261D/E370R/E547R, termed Ssb-3*), which significantly reduce indirect ribosome association of Ssb via Ssz1, likewise did not affect yeast growth under normal conditions[28].

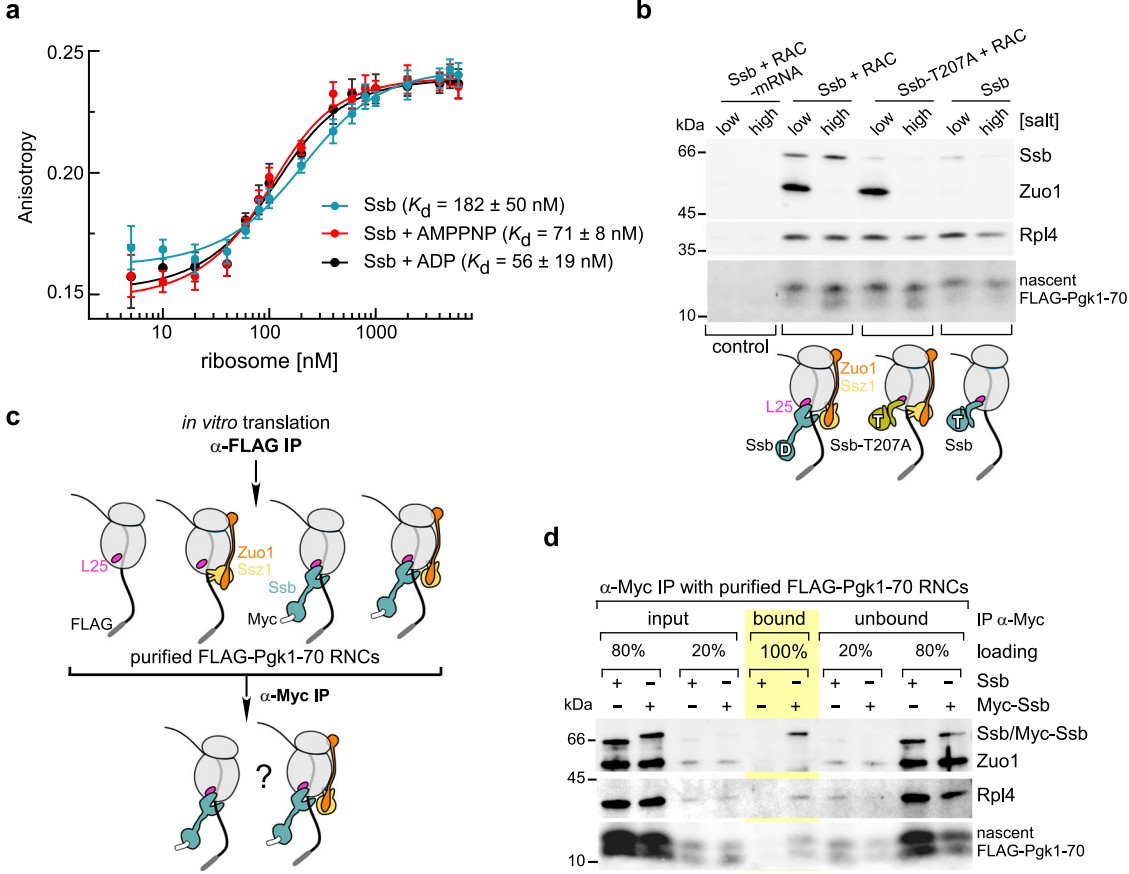

**Fig. 4 | Binding of Ssb to nascent chains enhances affinity/stability and salt-resistance of the Ssb·RNC complexes. a** Ssb binds to non-translating ribosomes with high affinity. Fluorescence anisotropy-based binding assays were performed with Ssb-ATTO647N (100 nM) and purified non-translating 80S ribosomes. Shown are Anisotropy binding curves of a single biological replicate, the center indicates the mean, error bars represent the standard deviation of five technical replicates. Dissociation constants ($K_d$) were determined with a 1:1 binding model based on 8 (Ssb) or 3 (Ssb + 1 mM AMPPNP, Ssb + 1 mM ADP) biological replicates (see Methods). **b** ATP hydrolysis by Ssb enhances stable, salt-resistant binding to RNCs. RNCs carrying nascent FLAG-Pgk1-70 in the presence of (Ssb + RAC), (Ssb-T207A + RAC), or Ssb alone were washed with low- or high-salt buffer and were subsequently isolated via FLAG-affinity purification. Purified RNCs (Rpl4), Ssb, RAC (Zuo1), and nascent FLAG-Pgk1-70 were detected by immunoblotting. The experiment was independently performed twice, yielding similar results. The cartoon indicates the nucleotide state of Ssb-ATP (T) or Ssb-ADP (D). RNCs (gray) Ssb (light teal) Ssb-

T207A (olive), Zuo1 (orange), Ssz1 (yellow), Rpl25 (light magenta). **c** Experimental approach to determine whether Ssb and RAC can simultaneously associate with the same RNC. RNCs carrying FLAG-Pgk1-70 bound to Myc-Ssb, or Ssb as a control, were subjected to two successive purification steps. First, RNCs were isolated via FLAG-affinity purification (IP α-FLAG), second, via Myc-affinity purification (IP α-Myc). Purified RNC·Myc-Ssb complexes were analyzed for the presence of RAC. Color code as in (**b**). **d** RAC is excluded from RNC·Myc-Ssb complexes. The experimental approach is outlined in **c**. The experiment was independently performed twice, yielding similar results. RNCs isolated by FLAG-affinity purification (input), Myc-affinity purified RNCs (bound), and supernatant of Myc-affinity purification (unbound) were analyzed by immunoblotting with the antibodies indicated, nascent FLAG-Pgk1-70 was detected with α-FLAG. Input and unbound were divided into 20% and 80% portions to achieve linearity of band intensities for all components with 100% bound.

We therefore asked whether a combination of the Ssb-DD1-DD2 with Ssb-3* mutations might result in synthetic growth defects. The Ssb-3* mutations caused moderate growth defects in our strain background (Fig. 3g and Supplementary Fig. 7d). Importantly, growth defects were significantly enhanced in a strain expressing Ssb-3*-DD1-DD2, which combines the two sets of mutations (Fig. 3g and Supplementary Fig. 7d). The synthetic growth defect suggested that proper function of the RAC-Ssb cycle was dependent on positioning of Ssb-ATP at the ribosome, either through the interaction of Ssb with Rpl25 or through interaction between the NBDs of Ssb and Ssz1. To test this possibility, ribosome-binding of Ssb-DD1-DD2 was compared with that of the Ssb-3*-DD1-DD2 mutant and with Ssb-DD1-DD2 expressed in the Δssz1 background (Fig. 3h and Supplementary Fig. 7e). According to the dual binding site model, the latter two mutants should exhibit markedly reduced binding. Indeed, ribosome-binding of Ssb-3*-DD1-DD2 and of Ssb-DD1-DD2 expressed in the Δssz1 background was further reduced by more than fivefold (Fig. 3h and Supplementary Fig. 7e) compared to the already strongly reduced ribosome-binding of Ssb-DD1-DD2

(Fig. 3b,c). These findings demonstrate that residual ribosome-binding observed for Ssb-DD1-DD2 is mediated by the interaction between the Ssb-NBD and Ssz1.

## Ssb binds to non-translating ribosomes with high-affinity, which is further enhanced when Ssb interacts with a nascent chain

To quantitatively characterize Ssb-ribosome interactions, we measured binding affinities with ATTO647N-labeled Ssb (Supplementary Fig. 8a,b). Fluorescence anisotropy changes followed a saturation binding curve, reflecting a single binding site with a $K_d$ of $182 \pm 50$ nM for Ssb in the absence of nucleotide (Fig. 4a). Addition of ADP or of the non-hydrolysable ATP analog AMPPNP resulted in a slight but significant affinity increase (Fig. 4a). We conclude that Ssb-ADP ($K_d$: $56 \pm 19$ nM) as well as Ssb-AMPPNP ($K_d$: $71 \pm 8$ nM) bind to non-translating ribosomes with similar, submicromolar affinity (Fig. 4a). To address the role of RAC in ribosome-binding of Ssb, we tested whether RAC affected the affinity of Ssb for non-translating ribosomes. To that end, 1:1 ribosome:RAC complexes were generated

(Supplementary Fig. 8c), and the binding of Ssb to ribosomes was compared with that to ribosome·RAC complexes (Supplementary Fig. 8d-f). The $K_d$ of Ssb for ribosome·RAC complexes was 163 ± 68 nM (Supplementary Fig. 8f), which resembled closely the $K_d$ of Ssb for ribosomes (Fig. 4a and Supplementary Fig. 8f). The data suggested that RAC does not enhance the affinity of Ssb for empty ribosomes, indicating that both ATP- and ADP-bound Ssb likely associate with non-translating ribosomes primarily through the interaction of the helix Ssb-αD with Rpl25, rather than through interaction of the NBDs of Ssb and Ssz1 (Fig. 3g,h). However, in cell extract-based systems, ribosome-binding of catalytically inactive Ssb, or wild type Ssb in the absence of RAC, was significantly reduced when compared to wild type Ssb in the presence of RAC (Supplementary Fig. 8g, h and[5]).

Since the interaction of Ssb with nascent chains depends on ATP hydrolysis, and RAC stimulates ATP hydrolysis by Ssb[11,21], the most plausible explanation for this discrepancy was that the interaction of Ssb with nascent chains established an additional binding site, enhancing the overall affinity of Ssb for ribosomal complexes. To directly test this hypothesis, we analyzed the relative occupancy of Ssb on FLAG-Pgk1-70 RNCs, to which Ssb binds efficiently (Supplementary Fig. 1d). In the presence of RAC, wild type Ssb was bound to RNCs in a salt-resistant manner (Fig. 4b). In the absence of RAC, significantly less Ssb was bound, and Ssb was released from RNCs upon high-salt treatment (Fig. 4b). RNC-binding of the ATPase-deficient Ssb-T207A mutant in the presence of RAC resembled RNC-binding of wild type Ssb without RAC. The data show that ATP-hydrolysis was required for efficient and salt-resistant binding of Ssb to RNCs. We conclude that RAC, by stimulating Ssb´s ATPase activity and allowing Ssb to interact with nascent chains, enhanced the binding affinity and salt-resistance of Ssb.

To understand how these interactions are coordinated structurally, the model of ribosome-bound Ssb in the ATP conformation and the atomic models Ssb-ADP S1 and S2 were overlaid with the atomic model of RAC C2 (PDB 7X3K[16],) (Supplementary Fig. 9a–f). Notably, in the model of ATP-bound Ssb, the NBDs of Ssb and Ssz1 were apart, however, faced each other in a way that was compatible with their interaction (Supplementary Fig. 9b, and see Discussion). When Ssb-ADP S1 was overlaid with RAC C2, Ssb-SBDβ was in close proximity of the Ssz1-NBD (Supplementary Fig. 9d), and in the case of Ssb-ADP S2, Ssb-SBDβ overlapped with the Ssz1-NBD atomic model (Supplementary Fig. 9f). These models suggested that RAC can associate with ribosomes bound to Ssb-ATP; however, RAC sterically clashes with Ssb-ADP, particularly in the S2 state. We thus examined whether RAC was bound to FLAG-Pgk1-70 RNCs associated with Ssb-ADP using a two-step purification protocol to isolate RNCs directly associated with Ssb (Fig. 4c). As expected, both Ssb and RAC were present in the initial FLAG affinity-purified RNCs (Fig. 4b and Fig. 4d input). However, following immunoprecipitation of Ssb-bound RNCs, RAC levels fell below the detection limit (Fig. 4d, α-Myc IP). Thus, translating ribosomes carrying a nascent chain bound to Ssb-ADP were no longer stably associated with RAC. We conclude that RAC-mediated ATP hydrolysis by Ssb not only promoted stable engagement with the nascent chain, but also led to RAC release, thereby defining a key mechanistic step in the cotranslational chaperone cycle.

## Discussion

Chaperones capture nascent chains as soon as they emerge from the ribosomal tunnel, tightly coupling protein synthesis to downstream steps of protein biogenesis[2,3,6]. This coupling comes at a cost: folding assistants operate under spatial and temporal constraints imposed by translation, as their substrates elongate rapidly[41,42] yet remain tethered to the peptidyl transferase center and are therefore not equally accessible. For Hsp70s, these restrictions are amplified by their own reaction cycle. They first bind in an open conformation, which, upon ATP hydrolysis, switches into a closed, high-affinity state

(Supplementary Fig. 1a and Introduction). Crucially, this conformational change occurs while the SBDβ substrate binding cleft remains at the ribosomal tunnel exit, potentially generating steric interference between Hsp70 domains and the ribosome. What seems a simple binding step thus becomes a precision event, requiring folding machineries adapted to the geometry and dynamics of translation.

The yeast RAC/Ssb system exemplifies the specialization required to meet the above-mentioned demands of cotranslational assistance. Progress toward a comprehensive understanding of the system has long been hindered by the lack of molecular details on Ssb-ribosome interactions[2,5,6,17,28]. In this study, we close this gap by providing structural and biochemical evidence demonstrating that Ssb-ATP as well as Ssb-ADP bind to Rpl25 via helix Ssb-αD (Fig. 2a, Supplementary Fig. 6a, Fig. 3e,f). Furthermore, we show that to ensure proper coordination within the RAC-Ssb chaperone system ribosome-binding of Ssb-ATP to Rpl25 (Fig. 3e,f) acts cooperatively with binding of Ssb-ATP to Ssz1 (Fig. 3g,h). On the basis of our data, and in consideration of previous observations, we propose a model (Fig. 5) that provides a consistent explanation for how the RAC-Ssb system fulfills the spatial and temporal constraints described above. We would like to note, however, that parallel pathways may exist, and that the properties of the nascent chain may influence the sequence of events during the RAC-Ssb cycle. The ensembles of ribosome-bound Ssb and RAC used to construct the model in Fig. 5 are detailed in Supplementary Note 2 and Supplementary Figs. 9 and 10.

The cycle initiates on ribosomes (Fig. 5 stage 1) to which Ssb-ATP (Fig. 5 stage 2 and Supplementary Fig. 9a) and RAC (Fig. 5 stage 3 and Supplementary Fig. 9b) bind independently yet concurrently. In stages 2 and 3 Ssb-ATP is bound via the contact of Ssb-αD with Rpl25 (Fig. 3e,f) and Ssb-SBDβ points away from the tunnel exit. RAC is bound such that the peptide-binding cleft of Ssz1-SBDβ is positioned in proximity of the tunnel exit (Fig. 5, stage 3 and Supplementary Fig. 9b). In this ensemble, Ssz1-SBDβ can receive the emerging nascent chain and the Ssz1-NBD aligns with the Ssb-NBD, facilitating formation of the Ssb-NBD/Ssz1-NBD complex in the next stages of the cycle (Fig. 5, stage 3, Supplementary Movie 1, and see below). Structural information on the following, likely transient, step of the cycle is not yet available. However, when supplied with Ssb, Zuo1, and Ssz1, AlphaFold 3 (AF3) predicts a complex, termed Ssb-ATP·RAC (Supplementary Fig. 10a–c), which can be positioned on the ribosome according to the established interaction of the Zuo1-ZHD with the large ribosomal subunit[15–17] (Fig. 5 stage 4, and Supplementary Fig. 10d,e). This Ssb-ATP·RAC complex fulfills the defining criteria of a ribosome-bound pre-hydrolysis complex: Ssb is in the open, ATP-bound conformation, Ssb-SBDβ is close to the tunnel exit, and the Zuo1-JD is oriented to stimulate ATP hydrolysis by Ssb (Fig. 5, stage 4). Remarkably, the Ssb-ATP·RAC complex can emerge from the RAC/Ssb-ATP ensemble depicted in stage 3, when the two NBDs come into contact and the Zuo1-ND unfolds (Supplementary Movies 2 and 3). Large conformational changes within the Zuo1-ND may at first seem unlikely, as structural and functional data indicate that it is deeply intertwined with Ssz1-SBDβ[18–20]. However, structural studies have demonstrated that ribosome-bound Zuo1 exhibits marked conformational plasticity[16,17]. We propose that the previously observed displacement of the Zuo1-LP pseudo-substrate from Ssz1-SBDβ by the nascent chain[21] contributes to the destabilization of the Zuo1-ND·Ssz1-SBDβ interaction, thereby promoting increased flexibility of the Zuo1-ND.

When ATP hydrolysis occurs in the Ssb-ATP·RAC complex Ssb undergoes pronounced conformational changes[40,43] that cannot be compensated by spatial rearrangements of Ssb-SBDβ, which is attached to the nascent chain (Fig. 5, transition from stage 4 to 5, Supplementary Movie 4). Our model predicts that SBDβ rotates approximately 90°, yet its distance from the tunnel exit remains largely unchanged. Ssb-SBDα, however, undergoes a pronounced movement, including an approximately 270° rotation relative to the

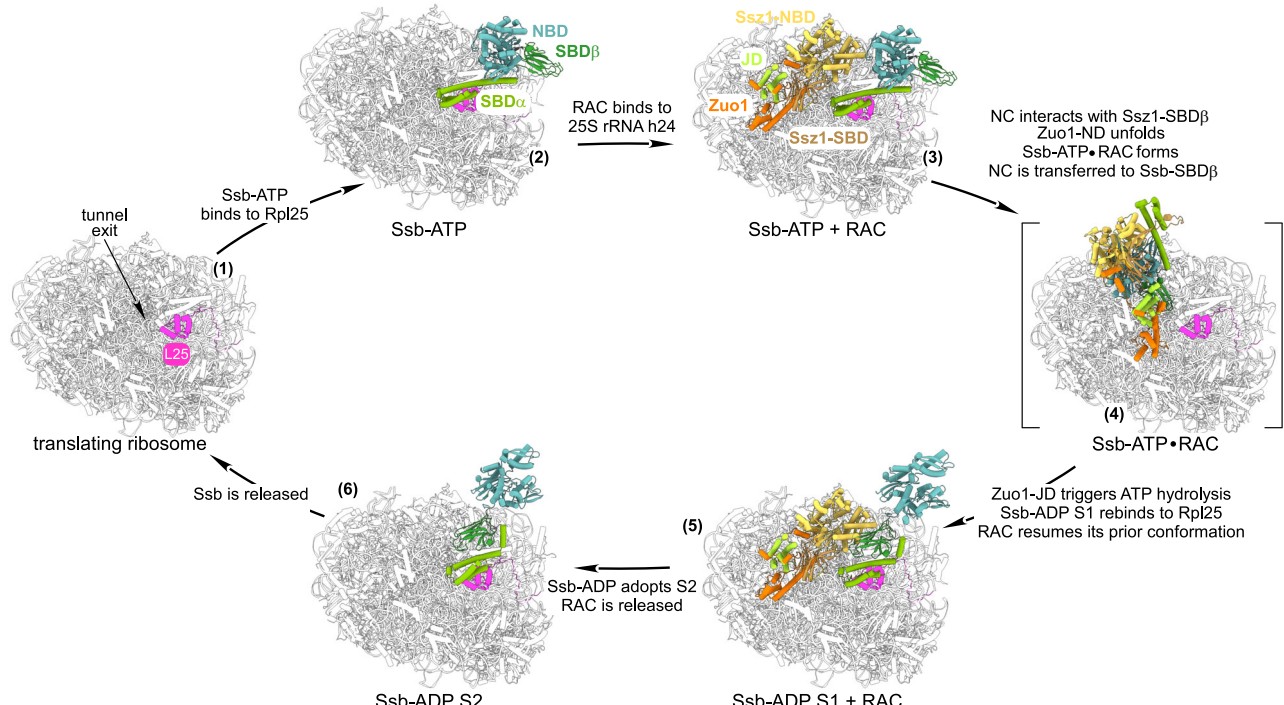

**Fig. 5 | Model of the full RAC-Ssb chaperone cycle on translating ribosomes.** The ribosomal protein Rpl25 (stage 1), serves as the primary binding site for Ssb-ATP (stage 2, Fig. 3e,f). h24 of the 25S rRNA serves as a primary binding site for RAC[15,16] (stage 3). At stage 3, Ssb-ATP and RAC bind without steric clashes; the Ssz1-SBDβ is near the tunnel exit[16], and the NBDs of Ssb and Ssz1 are prepositioned for complex formation (Supplementary Movie 1). The emerging nascent chain first binds to Ssz1-SBDβ, displacing the Zuo1-LP motif from Ssz1-SBDβ[21]. Freed from its contact with Ssz1, the Zuo1-ND exhibits significant flexibility[16]. This allows coordinated conformational changes leading to the formation of the Ssb-ATP·RAC complex, in which Ssb-ATP interacts with Ssz1[28]. The Zuo1-ND extends and the Ssb-ATP·RAC complex rotates along the ribosomal surface, leading to positioning of Ssb-SBDβ at tunnel exit and handover of the nascent chain from Ssz1-SBDβ to Ssb-SBDβ[21] (Supplementary Movie 2). In the pre-hydrolysis state, the Zuo1-ND is fully extended, and the Zuo1-JD is positioned to trigger ATP hydrolysis by Ssb (stage 4 and

Supplementary Movie 3). ATP hydrolysis triggers conformational changes within Ssb[40,43] that disrupt its interaction with the Ssz1-NBD and enable RAC to return to its prior conformation (stage 5 and Supplementary Movie 4). In stage 5, Ssb-ADP S1 is bound to the nascent chain (Fig. 1d) and has regained its contact with Rpl25 (Fig. 2a). The contact with Rpl25 is maintained as Ssb-ADP adopts the S2 conformation (Supplementary Fig. 6a). Ssb-SBDβ is now closer to the ribosomal surface and clashes with Ssz1-NBD, which leads to RAC release (stage 6, Supplementary Movie 5). Ssb-ATP (stage 2 and 3): Supplementary Fig. 9a; RAC (stage 3 and 5): PDB 7X3K[16]; Ssb-ADP S1 (stage 5): Fig. 1b, Ssb-ADP S2 (stage 6): Fig. 1b, Ssb-ATP·RAC (stage 4): AF3-predicted model (Supplementary Note 2, Supplementary Fig. 10, and Methods). Large ribosomal subunit[64] (PDB 6T7I, light gray), Zuo1 (orange), Zuo1-JD (limon), Ssz1-NBD (yellow orange), Ssz1-SBDβ (sand), Ssb-NBD (light teal), Ssb-SBDβ (forest) Ssb-SBDα (split pea), Rpl25 (light magenta).

ribosome, which relocates helix Ssb-αD to its binding site on Rpl25 adopting the Ssb-ADP S1 conformation (Fig. 5 stage 5, Supplementary Fig. 9d, Supplementary Movie 4). Please note that without RAC-mediated positioning, Ssb-ATP could engage nascent chains near the tunnel exit in multiple orientations, many of which would lead to steric clashes between Ssb-SBDα and the ribosome upon ATP hydrolysis. With regard to RAC, after transfer of the nascent chain to Ssb-SBDβ, the Zuo1-LP reassociates with Ssz1-SBDβ, allowing RAC to resume the RAC C2 conformation, which places the Ssz1-NBD in close proximity of ADP-bound Ssb-SBDβ (Fig. 5, stage 5, Supplementary Fig. 9d, Supplementary Movie 4). When Ssb-ADP adopts the S2 state RAC can no longer be accommodated, is released from the ribosome, and available for a next round of the chaperone cycle (Fig. 5 stage 5-6, Supplementary Fig. 9f, Fig. 4c,d, Supplementary Movie 5). The mechanism of Ssb release remains unclear at this point. Our data suggest two mutually non-exclusive possibilities. First, since Ssb binds nascent chains in a salt-resistant manner even in the absence of direct ribosome interaction (Fig. 3b,c), Ssb-ADP may dissociate from the ribosome while remaining bound to the elongating nascent chain. Alternatively, Ssb-ADP might remain associated with the ribosome for a prolonged period, allowing a nascent chain domain to fold before a nucleotide exchange factor triggers nucleotide exchange. In this scenario, Ssb, now in the open ATP-bound

conformation, could maintain its ribosome association via Ssb-αD, ready for the next RAC-mediated chaperoning cycle.

The model reconciles seemingly incompatible data regarding ribosome-binding of ATP-bound Ssb[5,28]. Ssb-ATP binds to ribosomes directly and with high affinity (Fig. 3e,f and Fig. 4a). Yet, during the cycle, Ssb-ATP is released from its ribosomal binding site and associates with Ssz1 in a way consistent with previous crosslinking data[28]. Such a dual recruitment likely ensures the robustness of cotranslational Hsp70 function. This is also supported with the observation that Ssb mutants that fail to interact with Rpl25 (Supplementary Fig. 7a[5,7]), as well as those that fail to interact with the Ssz1-NBD, display rather minor growth defects (Fig. 3g) and[28], whereas loss of both interactions results in pronounced growth defects (Fig. 3g) and failure of ribosome-binding (Fig. 3h). It is interesting to recall that HSPA1, the mammalian cotranslational Hsp70[24,44], does not directly interact with the ribosome via its SBDα[24]. Thus, HSPA1 might be recruited to ribosomes exclusively by complex formation with the NBD of the mammalian Ssz1 homolog termed HSPA14[22], or, alternatively, mammalian cells may have developed other safeguard mechanisms to assure proper positioning of the HSPA1-SBDβ close to the exit of the ribosomal tunnel. In essence, the proposed functional RAC-Ssb cycle (Fig. 5) highlights the reason for the remarkable complexity of JD partners associated with cotranslational Hsp70s in yeast and mammals. RAC not only

orchestrates timely ATP hydrolysis but also employs an intricate mechanism to position the SBDβ of its Hsp70 partner such that it can receive the nascent chain from Ssz1/HSPA14[21] in close proximity to the tunnel exit (Fig. 1d and[9,21]) and remain in this position when it hydrolyzes ATP. These findings are crucial for our understanding of the molecular mechanism underlying cotranslational protein folding and proteostasis in eukaryotic cells.

# Methods

## Media and general growth conditions

Yeast cells were grown at 30 °C in YPD medium (1% yeast extract, 2% peptone, 2% glucose, 20 µg ml$^{-1}$ adenine hemisulfate) and were harvested in log-phase. To induce translational run-off for the purification of non-translating ribosomes, log-phase cells were transferred to 30 °C YP medium (1% yeast extract, 2% peptone) followed by 10 min incubation 30 °C and 200 rpm[45]. E. coli cells for protein expression were grown in LB medium (0.5% yeast extract, 1% peptone, 1% NaCl) on a shaker at 110 rpm at the temperatures detailed below.

## Yeast strains

*Saccharomyces cerevisiae* (*S.c.*) strain MH272-3f is the parental strain of mutant strains employed in this study[46,47]. Genomic mutations R596D-K597D-K603D-R604D within *SSB1* and *SSB2* (Ssb-DD1-DD2 strain) and T207A (Ssb-T207A strain) were generated by CRISPR-Cas9 genome editing[48]. Gene deletions were generated by genome integration of PCR-amplified DNA fragments containing a selectable marker flanked by 300-500 bp up- and down-stream of the target gene. Ssb-T207A and Ssb-DD1-DD2 (with the exception of the strains shown in Fig. 3g,h and related to Supplementary Fig. 7d,e) were integrated into the genome (see below). All other mutated proteins were expressed from plasmids in the respective deletion background. Plasmids, yeast strains, and primer sequences are listed in Supplementary Tables 2-4.

## Generation of the genomic Ssb-T207A and Ssb-DD1-DD2 strains

CRISPR-edited clones were selected based on resistance to nourseothricin (pRCC-N) or geneticin (pRCC-K)[48]. Genomic modifications were verified by sequencing. Loss of pRCC-K and pRCC-N plasmids was by growth of yeast strains on antibiotic-free YPD medium. Primer and single strand donor DNA for CRISPR-Cas9 (dDNA) sequences are listed in Supplementary Table 4. Ssb-T207A strain (Supplementary Table 3) pRCC-K_SSB1/SSB2_PAM634 was constructed by Gibson assembly (GeneArt, Invitrogen)[49] of PCR products generated with primer pairs WGP235-pCC2_Fw/CC_Ssb_PAM634 fw and WGP234-pCC1_Rv/CC_Ssb_PAM634 rev and pRCC-K as a template. MH272-3fα was co-transformed with pRCC-K_SSB1/SSB2_PAM634 and dDNA_Ssb-T207A_HindIII. Yeast clones carrying the T207A mutation in *SSB1* and SS*B2* (Ssb1-T207A and Ssb2-T207A) were selected based on geneticin resistance and slow growth displayed by strains expressing Ssb1-T207A and Ssb2-T207A (Supplementary Fig. 7a). Due to sequence variation in the 3´-region of the *SSB1* and *SSB2 orfs*, the R596D-K597D (DD1) mutation was introduced into *SSB1* and *SSB2* successively. First, the DD1 mutation was introduced into *SSB2*. To that end, pRCC-N_SSB2_PAM1789 was generated with primer pair WGP235-pCC2_Fw/CC_Ssb2 PAM1789 fwd and WGP234-pCC1_Rv/CC_SSB2 PAM1789 rev using pRCC-N as a template. pRCC-N_SSB2_PAM1789 was co-transformed with dDNA Ssb2_DD1 resulting in the strain Ssb2-DD1. Second, the DD1 mutation was introduced into *SSB1* using pRCC-K SSB1_PAM1789 generated with primer pair WGP235-pCC2_Fw/CC_Ssb1 PAM 1789 fwd and WGP234-pCC1_Rv/CC_Ssb1 PAM 1789 rev using pRCC-K as a template. pRCC-K SSB1_PAM1789 was co-transformed with dDNA Ssb1/Ssb2_DD1 into Ssb2-DD1, resulting in strain Ssb-DD1. The K603D-R604D (DD2) mutation was then introduced into *SSB1-DD1* and *SSB2-DD1*. To that end, pRCC-K_SSB1/SSB2_PAM1823 was generated with primer pair WGP235-pCC2_Fw/CC_Ssb1/2-PAM1823 fwd and WGP234-pCC1_Rv/CC_Ssb1/2 PAM 1823 rev using pRCC-K as a template. pRCC-K_SSB1/SSB2_PAM1823 was co-transformed with dDNA SSB1-2_DD1_DD2_AatII into Ssb-DD1. The resulting strain, termed Ssb-DD1-DD2, carries the R596D-K597D-K603D-R604D quadruple mutation in *SSB1* and *SSB2*.

## Plasmid construction

Primer and synthetic DNA sequences (Eurofins Genomics) for the construction of yeast vectors[50] are listed in Supplementary Table 4. pYEplac112-Rpl25 was constructed by transfer of *RPL25* ± 300 (primer pair Rpl25-300-PstI-F/Rpl25-300-BamHI-R) into pYEplac112[50]. pYEplac112-Rpl25-AAA and pYEplac112-Rpl25-KKK were generated by replacing the StyI/PacI fragment within pYEplac112-Rpl25 with synthetic DNA encoding for the corresponding StyI/PacI fragments encoding for *RPL25*-E77A-D131A-D134A or *RPL25*-E77K-D131K-D134K, respectively (Supplementary Table 4). For the construction of Ssb1 mutants *SSB1* ± 300 bp (primer pair Ssb1-BamHI-F/Ssb1-PstI-R) was introduced into pSP65 (Promega). For the generation of the single cysteine mutant, the ClaI/HindIII fragment within *SSB1* was replaced with a synthetic DNA fragment encoding for *SSB1*-C454S-C435S-C20S (Ssb1-C454S-C435S-C20S, Supplementary Table 4) resulting in pSP65-Ssb1-C454S-C435S-C20S. The HindIII/XbaI DNA fragment within the *SSB1*-C454S-C435S-C20S DNA sequence was then replaced with a synthetic DNA fragment encoding for the D589C mutation (Ssb1-D589C, Supplementary Table 4), resulting in pSP65-Ssb1-C454S-C435S-C20S-D589C. *SSB1*-C454S-C435S-C20S ± 300 bp and *SSB1*-C454S-C435S-C20S-D589C ± 300 bp were transferred into the BamH1/Pst1 site of pYCplac33, resulting in pYCplac33-Ssb1-C3S and Ssb1-C3S-D589C, respectively, which were expressed in the Δ*ssb1*Δ*ssb2* background (Supplementary Table 2). The resulting Ssb1-C3S and Ssb1-C3S-D589C strains display wild type growth (Supplementary Fig. 8a). pET28a (Novagen) was employed for the expression of N-terminally His$_6$-tagged wild type, or mutant versions of Ssb1 in E. coli. For that purpose, *SSB1-T207A* was amplified from Ssb-T207A genomic DNA, *SSB1-DD1-DD2* from Ssb-DD1-DD2 genomic DNA, and *SSB1-C3S-D589C* from pYCplac33-Ssb1-C3S-D589C with primer pair Ssb1-NdeI-F1/Ssb1-BamHI-R1. Each PCR product was cloned into pET28a. The resulting plasmids are termed pET28a-His$_6$-Ssb1-T207A, pET28a-His$_6$-Ssb1-DD1-DD2, and pET28a-His$_6$-Ssb1-C3C-D589 (Supplementary Table 1). For the expression of RAC, the *SSZ1 orf* was transferred into the NdeI/XhoI site of pET28N[18] resulting in pET28N-His$_6$-Ssz1, which was co-transformed with pETcoco2-Zuo1[18]. Dap2 ± 300 bp carrying a C-terminal 3x HA-tag was cloned into the PstI/BamHI site of pYEplac195, resulting in pYEplac195-Dap2-3HA. Low copy plasmids for the expression of Ssb1 mutants in the Δ*ssb1*Δ*ssb2* background are based on pYCplac-33-Ssb1[18]. pYCplac33-Ssb1-3* and pYCplac33-Ssb1-3*-DD1-DD2 were generated by replacing the AleI/PstI fragment of pYCplac-33-Ssb1 with synthetic DNA Ssb1-R261D-E370R-E547R (Supplementary Table 4) or Ssb1-R261D-E370R-E547R-R596D-K597D-K603D-R604D (Supplementary Table 4), respectively. In addition to the mutations within Ssb1, these synthetic DNAs introduced a PacI site 3´ of the Ssb1 stop codon (Supplementary Table 4). pYCPlac33-Ssb1-DD1-DD2 was constructed by replacing the AleI/PacI fragment of pYCPlac33-Ssb1-3*-DD1-DD2 with a PCR product generated with the primer pair Ssb1-AleI-F/Ssb1-PacI-R (Supplementary Table 4) using pET28a-His$_6$-Ssb1-DD1-DD2 as a template. pYCPlac33-Ssb1-DD1-DD1, pYCPlac33-Ssb1-3*, and pYCPlac33-Ssb1-3*-DD1-DD2 were transformed into Δ*ssb1*Δ*ssb2*[51] and pYCPlac33-Ssb1-DD1-DD2 was transformed into Δ*ssb1*Δ*ssb2*Δ*ssz1*. The resulting strains are termed Ssb1-DD1-DD2, Ssb1-3*-DD1-DD2, and Δ*ssz1* Ssb1-DD1-DD2.

## In vitro transcription and translation

DNA templates for transcription reactions were PCR-amplified using pSPUTK-Dap2, pSPUTK-Pgk1 or pSPUTK-FLAG-Pgk1[37] as a template. Reverse primers (Supplementary Table 4) were designed such that PCR

products encoded stop codon-less FLAG-Pgk1-40, FLAG-Pgk1-50, FLAG-Pgk1-60, FLAG-Pgk1-70, FLAG-Pgk1-80, FLAG-Pgk1-87, FLAG-Pgk1-100, Pgk1-120, and Dap2-120, respectively. Pgk1 and Dap2 transcripts were generated using SP6 polymerase (Thermo Fisher Scientific)[52]. Yeast translation extracts were prepared from wild type, $\Delta ssb1\Delta ssb2$, $\Delta ssb1\Delta ssb2\Delta zuo1\Delta ssz1$, and $\Delta srp54$[52]. Ribosome nascent chain complexes (RNCs) were generated using yeast translation extracts[52] primed with stop codon-less FLAG-Pgk1, Pgk1, or Dap2 transcripts of the length indicated in the Figure Legends[21,53]. Translation reactions were performed with 35% yeast translation extract in translation buffer (1.0 mM ATP, 0.08 mM GTP, 17.5 mM creatine phosphate, 0.2 mg/ml creatine phosphokinase, 0.2 mM of each amino acid, 2.2 mM putrescine, 0.1 mg/ml yeast tRNA, 100 U/ml RiboLock (Thermo Fisher Scientific)) at 20 °C as previously described[53,54]. After 80 min incubation at 20 °C translation reactions were stopped by addition of cycloheximide (final concentration of 50 μg ml-1). Ribosomal proteins, ribosome-associated factors, and FLAG-tagged nascent chains were detected by immunoblotting using the antibodies indicated in the Figures. In some experiments, RNCs were generated in the presence of [35S]-methionine (Hartmann-Analytic) to enable radiolabel-based visualization of the nascent chain. In these experiments unlabeled methionine was excluded from the translation reaction.

## Chemical crosslinking and identification of crosslink products by denaturing immunoprecipitation

RNCs carrying nascent Dap2-120 or Pgk1-120 were generated as described above, were isolated by centrifugation, and were resuspended in ribosome-binding buffer (20 mM HEPES-KOH, 2 mM Mg(OAc)$_2$, 120 mM KOAc, 50 μg ml-1 cycloheximide, 2 mM dithiothreitol, 1 mM PMSF (phenylmethylsulfonyl fluoride), 1x PIM (protease inhibitor mix), pH 7.4). Crosslinking was performed by addition of the amino-reactive, homo-bifunctional crosslinker bis-(sulfosuccinimidyl)-suberate (BS$^3$, Thermo Scientific), which was added to a final concentration of 400 μM. Immunoprecipitation with crosslinked samples was performed under denaturing conditions using protein A-Sepharose beads (CL-4B; GE Healthcare) pre-coated with α-Ssb[37].

## FLAG- and Myc-affinity purification of RNCs

For FLAG-affinity purification (Fig. 4b), RNCs containing FLAG-tagged Pgk1-70 were generated in translation extracts derived from either $\Delta ssb1\Delta ssb2$ or $\Delta ssb1\Delta ssb2\Delta zuo1\Delta ssz1$ strains. Translation reactions were layered onto either low-salt (25% w/v sucrose in ribosome-binding buffer) or high-salt (25% w/v sucrose in ribosome-binding buffer supplemented with 800 mM KOAc) sucrose cushions and were centrifuged at 400,000 x g for 35 minutes at 4 °C. Ribosomal pellets containing RNCs were resuspended in IP buffer (20 mM HEPES-KOH, 2 mM Mg(OAc)$_2$, 120 mM KOAc, 2 mM dithiothreitol, 50 μg ml-1 cycloheximide, 1 mM PMSF, 1x PIM, pH 7.4). Where indicated (Fig. 4b), purified His$_6$-Ssb or His$_6$-Ssb-T207A was added to a final concentration of 1 μM, followed by FLAG-affinity purification using anti-FLAG M2 affinity gel (Sigma)[55]. For the two-step purification (Fig. 4c), FLAG-Pgk1-70 mRNA was translated in $\Delta ssb1\Delta ssb2$ extract for 80 minutes at 20 °C. Translation reactions were then supplemented with ribosome-free total cell extract from either mycSsb1-expressing cells or wild type (control), prepared by centrifugation of total extract at 400,000 x g for 35 minutes. After a 10-minute incubation at 20 °C, FLAG-Pgk1-70 RNCs were isolated using FLAG-beads. Subsequently, FLAG-beads were washed with IP buffer and FLAG-Pgk1-70 RNCs were eluted with FLAG peptide (200 μg ml⁻¹ in IP buffer). To isolate FLAG-Pgk1-70 RNCs associated with mycSsb1, eluted FLAG-Pgk1-70 RNCs were subjected to a second affinity purification using anti-Myc agarose beads (Thermo Scientific). After washing with IP buffer, Myc-beads were boiled in SDS sample buffer and were analyzed by immunoblotting with the antibodies indicated in Fig. 4b, d.

## Purification of RNCs and preparation of cryo-EM grids

RNCs carrying FLAG-Pgk1-70 were generated in $\Delta ssb1\Delta ssb2$ extract complemented with purified, untagged Ssb1, or complemented with untagged Ssb and His$_6$-Snl1-ΔTM (Supplementary Fig. 6f) each at a final concentration of 3 μM. His$_6$-Snl1-ΔTM was included in the translation reaction because it was previously suggested that the BAG-domain protein Snl1, lacking its single transmembrane domain (Snl1-ΔTM) facilitates the recruitment of Ssb to translating ribosomes[56]. For the purification of FLAG-Pgk1-70 RNCs for cryo-EM analysis, translation reactions were diluted 1:5 with IP-buffer, RNCs carrying FLAG-Pgk1-70 were affinity purified using FLAG-beads, and were subsequently released with FLAG-peptide (see above and[55]). Four μl of released RNCs FLAG-Pgk1-70 (RNC concentration ~ 50 nM) were incubated on R3.5/1 200 mesh copper grids (Quantifoil) prefloated with 2 nm carbon for 30 s, were blotted for 1 s, and were subsequently plunge frozen in liquid ethane using an EM-GP2 plunge-freezer (Leica). The temperature in the loading chamber was set to 4 °C and the humidity in the loading chamber was set to 75%.

## Data collection and image processing

Cryo-EM data were collected on grids with FLAG-Pgk1-70 RNCs prepared in the presence of Ssb, with or without His$_6$-Snl1-ΔTM. Initial reconstructions revealed Ssb density near the tunnel exit in both samples, while no density for His$_6$-Snl1-ΔTM was observed. The sample containing His$_6$-Snl1-ΔTM showed more clearly defined Ssb density, which could potentially indicate a positive effect of Snl1-ΔTM on Ssb recruitment or positioning at the tunnel exit. However, we consider it more likely that this variation reflects the intrinsic variability of cryo-EM data obtained from affinity purified, in vitro generated RNCs, as well as differences in cryo-EM sample preparation, grid handling, and data collection. As more defined Ssb density provided an advantage for subsequent data collection, the His$_6$-Snl1-ΔTM dataset was selected for further analysis. Movies were collected using a 300 kV Titan Krios G4 equipped with a cold field emission gun, a Selectris energy filter (Thermo Fisher Scientific) with 10 eV slit width and a Falcon 4i direct electron detector. Micrographs were acquired at 130,000 x magnification (pixel size 0.951 Å/px) and a cumulative fluence of 40 e/Å$^2$ using a defocus range between -1.0 to -2.0 μm in 0.25 μm increments in EER format (40x fractionated)[57]. Data collection was automated using EPU (Thermo Fisher Scientific). The 9995, 10476 and 36270 movies were patch motion corrected and CTF estimated in CryoSPARC live (v 4.2.1, 4.4.1 and 4.5.1) (v 4.2.1, 4.4.1 and 4.5.1)[58]. Defocus and CTF fit values were used to curate the exposures to 9606, 10471 and 34209 micrographs, respectively. Particle picking was performed using cryolo (v 1.9)[59] by refining pretrained weights using 10 to 15 manually picked micrographs. Extracted particles (180 px 2.64 Å/px) were separated into batches of around 225,000 particles and sorted in four rounds of 2D classifications using CryoSPARC (v.4.2.1 to 4.5.1). Ab initio reconstruction was performed on the first collection to obtain 80S and 60S reference volumes that were used for particle sorting in heterogeneous refinements. 80S particles were refined and used for cryoDRGN variational autoencoder training (128 px, 3.71 Å/px, 3 layers for encoder and decoder each, for 50 epochs in an 8-dimensional latent space)[60]. Using the cryoDRGN landscape' tool, latent space encodings were first clustered via kmeans (k = 1000) and the resulting reconstructions, generated by back projection, were further clustered via agglomerative clustering (k = 20) (https://ez-lab.gitbook.io/cryodrgn/). The volumes of the resulting clusters are shown as centroids, representing voxel-based averages of the reconstructions within each cluster (Supplementary Fig. 3a, b). Latent space dimensionality reduction was performed by uniform manifold approximate and projection (UMAP). The resulting reconstructions of ribosomal states across the translation cycle were analyzed for density near the ribosomal tunnel exit, the hypothesized binding site of Ssb. Best

resolved density was observed in the cluster representing non-rotated, post-translocation state ribosomes, with a P-site tRNA and a peripheral density corresponding to a trailing ribosome (Supplementary Fig. 3, Custer 1). Corresponding particles were reextracted (180 px boxsize, 2.64 Å/px), converted to relion star files using cs2star/pyem, (https://github.com/brisvag/cs2star)[61]. The particle stacks from the combined collections 1 and 2 as well as collection 3 were refined in relion (v 5.0)[62] followed by a focused 3D classification (10 classes, $T = 300$) masking around the Ssb density. The ellipsoid mask was generated in ChimeraX (v.1.7) using the 'shape' command followed by the 'volume onesmask' command. Subsequently, the volume was binarized and dilated by a 'soft-edge' (2 px, 2.64 Å/px) in CryoSPARC[58]. The particles from collection 1 + 2 were combined with particles from the collection 3 for each Ssb conformation (S1 and S2) followed by further 3D classification masking around the Ssb density in both states by using the eraser sphere tool followed by binarization and dilation in CryoSPARC[58]. Particles were refined and 3D classified (2 classes, target resolution 6 Å, forced hard classification) once more in CryoSPARC using previously generated masks, followed by another round of refinement and 3D classification focusing on the P-site tRNA using a mask generated with the volume eraser tool as previously described. Particles with more defined tRNA density were selected for homogeneous refinement. The resulting maps were post processed via DeepEMhancer using the 'high-res' weights provided by the developers[63] resulting in cryo-EM maps Ssb-ADP S1 and S2 (Fig. 1b). Local resolution estimation was performed with CryoSPARC[58].

### Model building

A non-rotated, post-translocation ribosome with P- and E-site tRNA PDB 6T7I[64] was fitted in our densities as a rigid body fit in chimera (v 1.7-1.8[65]) as an initial model, together with the AlphaFold2 prediction[66] of the Ssb-SBD. The tRNA was replaced by tRNA$^{Met-e}$. Next, models were refined manually using coot (v 0.9.8[67]) and Isolde (v 1.8[68]). The nascent chain was modeled de novo using coot. Subsequently, real space refinement was performed using phenix (v 1.21[69,70]), followed by assessment of model statistics by phenix validation[71,72]. Problematic regions were manually fixed and the model was refined in phenix iteratively.

### Structure prediction

The structures of full-length *S.c.* Ssb-ATP and *S.c.* Ssb-ADP were predicted by I-TASSER[73] using *Chaetomium thermophilum* (*C.t.*) Ssb-ATP (PDB 5TKY[5]) and *E. coli* DnaK-ADP (PDB 2KHO[29]) as a template. The complex consisting of Ssb-ATP, Zuo1, and Ssz1 was predicted by AF3[74] (Supplementary Fig. 10 and Supplementary Note 2). Generation of models for the RAC-Ssb chaperone cycle (Fig. 5) is detailed in Supplementary Note 2 and Supplementary Fig. 10. Animated models (Supplementary Movies 1-5) were created using the morph function in ChimeraX (1.8-1.9) and structures presented in Fig. 5. To avoid clashes, morphing was performed using manually created intermediate structures.

### Heterologous protein expression and purification

*E. coli* expression plasmids are listed in Supplementary Table 1. His$_6$-Ssb1, His$_6$-Ssb1-T207A, His$_6$-Ssb1-DD1-DD2, His$_6$-Ssb1-D589C, and His$_6$-Snl1-ΔTM were expressed in Rosetta (DE3) (Novagen). For the purification of RAC, His$_6$-Ssz1 and Zuo1 were co-expressed in *E. coli* BL21 (DE3). Expression was induced with 1 mM isopropyl-β-D-1-thiogalactopyranoside at an OD$_{600}$ of 0.5 followed by 4 hours incubation at 25 °C for the expression of His$_6$-Ssb1 and His$_6$-Ssb1-variants, 4 hours incubation at 26 °C for expression of RAC, and for 4 hours incubation at 37 °C for expression of His$_6$-Snl1-ΔTM. His-tagged proteins were purified using ÄKTA start (GE Healthcare) and a Ni$^{2+}$-NTA column (PureCube Ni-NTA Cartridge 5 ml, Cube Biotech). Elution of proteins bound to the Ni$^{2+}$-NTA column was with a 30 ml linear 50 - 500 mM imidazole gradient in NiP1-buffer (20 mM HEPES-KOH, 100 mM KCl, 2.5 mM Mg(OAc)$_2$, 1 mM PMSF, pH 7.8). RAC was purified by Ni$^{2+}$-NTA purification followed by a second purification step using anion-exchange chromatography on MonoQ-5/50 GL (GE Healthcare) with a 20 ml linear 50 - 500 mM NaCl gradient in 20 mM HEPES-KOH, pH 7.4. His$_6$-Snl1-ΔTM was purified by Ni$^{2+}$-purification followed by size-exclusion chromatography (ÄKTA pure, GE Healthcare) on Superdex 200 10/300 GL (GE Healthcare) equilibrated with NiP1-buffer adjusted to pH 7.4. Eluted proteins were transferred to NiP2 buffer (20 mM HEPES-KOH, 100 mM KCl, 2.5 mM Mg(OAc)$_2$, pH 7.4) with PD10 columns (GE Healthcare). Purified His$_6$-Ssb1 variants and RAC preparations are shown in Supplementary Fig. 6f. For the preparation of cryo-EM samples the His$_6$-tag was removed from purified His$_6$-Ssb1 by cleavage with thrombin (Thrombin CleanCleave, Sigma) in cleavage buffer (20 mM HEPES-KOH, 100 mM KCl, 2.5 mM Mg(OAc)$_2$, pH 7.8) at 20 °C overnight. After addition of 2 mM PMSF, thrombin agarose beads were removed by centrifugation at 500 x g and the supernatant containing untagged Ssb was clarified by centrifugation at 20,000 x g for 20 min at 4 °C, was transferred to NiP2 buffer containing 1x PIM as described above (Supplementary Fig. 6f).

### Purification of non-translating ribosomes

Ribosome-binding assays and Anisotropy experiments were performed with freshly prepared non-translating ribosomes. Example blots are shown in Supplementary Fig. 6d,e. Ssb-free non-translating ribosomes were obtained from the *myc*Ssb1-ΔC23 strain[5], which does not suffer from ribosome assembly defects observed in Δ*ssb1*Δ*ssb*2 strains[54]. Ribosomes carrying Rpl25-E77K-D131K-D134K were purified from the Rpl25-KKK strain. Typically, a 2l log-phase YPD culture was glucose-depleted as described above. After harvest, glucose-depleted cells were rapidly frozen in liquid nitrogen, and subsequently disrupted with a cryo-mill (MM400, Retsch). The cryo-mill powder was resuspended in high-salt lysis buffer (20 mM HEPES-KOH, 100 mM KOAc, 500 mM KCl, 2.5 mM Mg(OAc)$_2$, 1x PIM, 1 mM PMSF, 1 mg ml$^{-1}$ Heparin, 2 mM dithiothreitol, pH 7.4) at a ratio of 1 mg cryo-mill powder per 5 ml buffer. The resulting lysate was clarified by centrifugation at 20,000 x g for 10 min at 4 °C followed by a second centrifugation step at 40,000 x g for 20 min at 4 °C. Fifteen to 20 ml of the clarified supernatant was loaded onto a 2.5 ml high-salt sucrose cushion (25% sucrose (w/v) in high-salt lysis buffer) and ribosomes were collected by centrifugation at 400,000 x g at 4 °C in T865 rotor (Thermo Scientific). Ribosomes were resuspended in high-salt lysis buffer and were incubated for 30 min at 4 °C and 1000 rpm (IKA VXR Basic Vibrax Shaker) followed by a 2 min clarifying spin at 2,000 x g. Solubilized ribosomes (500 μl) were loaded onto a 500 μl high-salt sucrose cushion and ribosomes were collected by centrifugation for 30 min at 400,000 x g in a MLA-130 rotor (Beckman Coulter) at 4 °C. Ribosomes were resuspended in storage buffer (20 mM HEPES-KOH, 100 mM KCl, 2.5 mM Mg(OAc)$_2$, pH 7.4) and were incubated at 1,000 rpm 4 °C for 1 h followed by a 2 min clarifying spin at 2,000 x g 4 °C prior to use.

### Ribosome-binding assays

Ribosome-binding assays with total cell extract were performed with cycloheximide (final concentration 100 μg ml$^{-1}$) treated log-phase cells. Cells were resuspended in ribosome-binding buffer and were disrupted by the glass beads method[5]. After a clearing spin at 20,000 x g, each 60 μl of the total cell extract (A$_{260}$ between 90 and 100 mAu) was loaded onto a 90 μl low-salt sucrose cushion (25% (w/v) sucrose in ribosome-binding buffer or onto a 90 μl high-salt sucrose cushion (25% (w/v) sucrose in 20 mM HEPES-KOH, 800 mM KOAc, 2 mM Mg(OAc)$_2$, 50 μg ml$^{-1}$ cycloheximide, 2 mM dithiothreitol, 1 mM PMSF, 1x PIM, pH 7.4). After centrifugation at 400,000 x g for 35 min at 4 °C total cell extract, cytosolic supernatant and ribosomal pellets were analyzed as described[5]. For ribosome-binding assays with purified components

typically 2 μM non-translating ribosomes were incubated with 1 μM His$_6$-Ssb1 or His$_6$-Ssb1-variants in ribosome-binding buffer lacking cycloheximide for 2 min at 25 °C. After a clarifying spin at 2000 x g, 20 μl of the sample was loaded onto a 30 μl low-salt sucrose cushion. Ribosomes and bound Ssb were isolated by centrifugation at 400,000 x g (TLA100, Beckman Coulter) at 4 °C. Ribosome-binding assays in the presence of nucleotide were performed with purified His$_6$-Ssb or His$_6$-Ssb variants pre-incubated at 22 °C with 1 mM ADP or ATP for 2 min in ribosome-binding buffer containing 1 mM ATP or ADP as indicated.

## Anisotropy measurements

Purified His$_6$-Ssb1-C3S-D589C (100 - 200 μM) was incubated with a two-fold molar excess of Atto647N maleimide (ATTO-TEC) in NiP2 buffer with 1x PIM. The reaction was incubated overnight at 20 °C and was then quenched with dithiothreitol (1 mM final concentration). Labelling efficiency, determined by absorption spectroscopy (NanoDrop 2000, Peqlab) according to the ATTO-TEC protocol, was above 88% in all preparations (Supplementary Fig. 8b). Free dye was removed by gravity flow using PD G-25 MiniTrap columns (Cytiva) equilibrated with NiP2 buffer. Anisotropy measurements were performed in storage buffer using a plate reader (TECAN Spark, 28 °C, $\lambda_{ex}$ = 635 nm, $\lambda_{em}$ = 680 nm, 20 nm excitation and emission slit). The concentration of His$_6$-Ssb1-ATTO647N was held constant (100 nM) and increasing concentrations of non-translating ribosomes or 1:1 ribosome·RAC complexes (Supplementary Fig. 8c) was added. Ssb was pre-incubated with 1 mM ADP or AMPPNP (Sigma) as indicated prior to addition of ribosomes in NiP2 buffer, containing 1 mM ADP or AMPPNP, respectively. To determine dissociation constants, Anisotropy values were plotted as a function of increasing ribosome concentrations and were fitted to a 1:1 binding model with GraphPad Prism 10.1.1 using equation

$$r = r_{unbound}\left(1 - \frac{[SR]}{[S_0]}\right) + r_{bound}\frac{[SR]}{[S_0]} \qquad (1)$$

where [SR] was replaced by

$$[SR] = \frac{([S_0] + [R_0] + K_d) - \sqrt{([S_0] + [R_0] + K_d)^2 + 4 \cdot [S_0][R_0]}}{2} \qquad (2)$$

Anisotropy values were weighted by 1 over their variance. Eight biological replicates were performed to determine $K_d$ values of Ssb for non-translating ribosomes in the absence of nucleotide, 3 biological replicates were performed to determine $K_d$ value of Ssb for the ribosome·RAC complex, and 3 biological replicates were performed to determine $K_d$ values of Ssb for non-translating ribosomes in the presence of ADP or AMPPNP, respectively.

## Quantification and statistical analysis

Quantitative analysis of immunoblots was with Image J 1.54 g (National Institutes of Health, USA). Statistical analysis was performed with at least 3 biological replicates using GraphPad Prism 10.1.1. The number of replicates used for statistical analysis is indicated in the Figure Legends. Analysis of Variance (ANOVA) with Tukey´s test for multiple comparison was used to determine statistic significance. Statistical analysis for ribosome-binding assays was by Two-way ANOVA, with the exception of Supplementary Fig. 8e (paired t-test). p values are indicated in the Figures. p values > 0.05 are regarded as non significant.

## Antibodies

Rabbit polyclonal antibodies α-Ssb[21] (1:10,000, #471), α-Zuo1[53] (1:20,000, #10), α-Ssz1[53] (1:8,000, #11), α-Rpl35[5] (1:5,000, #99), α-Srp54[53] (1:2,000, #12), α-Sse1[53] (1:10,000, #31), α-Rpl25[55] (1:5,000, #101), α-Rpl4[75] (1:8,000, #167), α-Rps9[53] (1:5,000, #44), and α-Pgk1[75] (1:10,000, #407) (Eurogentec, Rospert lab antibody collection). α-His$_6$

(1:4,000, Biorad, 154295), α-FLAG-tag (1:2,000, Sigma, F1804), α-HA (1:2000, Invitrogen, 26183) are purified monoclonal antibodies. Horseradish peroxidase-conjugated protein A (1:5,000, Thermo Fisher Scientific, 101023) or goat anti Mouse HRP (1:10,000, Santa Cruz, sc-2748) was employed as a secondary antibody. Numbers in brackets indicate the antibody dilution for immunoblotting and the antibody collection number).

## Miscellaneous

PIM (proteinase inhibitor mix), 1000-fold concentrated in dimethyl sulfoxide: 1.25 mg ml$^{-1}$ leupeptin, 0.75 mg ml$^{-1}$ antipain, 0.25 mg ml$^{-1}$ chymostatin, 0.25 mg ml$^{-1}$ elastinal, 5 mg ml$^{-1}$ pepstatin A. For the determination of protein levels in yeast cells, lysates were generated under denaturing conditions[76]. Protein s samples were separated on 10% Tris-Tricine gels[77]. Immunoblotting of protein samples was onto PVDF (Amersham) or nitrocellulose (Amersham) membrane. Immunoblots were developed by enhanced chemiluminescence using an ECL camera[78]. Protein concentrations were determined with the Bradford assay[79], and in the case of recombinant Ssb1 with the calculated extinction coefficient at 280 nm ($\varepsilon_{Ssb1}$ = 18910 M$^{-1}$cm$^{-1}$). The concentration of purified ribosomes was determined using the extinction coefficient at 260 nm ($\varepsilon_{ribosome}$ = 6 ×107 M$^{-1}$ cm$^{-1}$)[80]. Autoradiographs and fluorescently labeled proteins were analyzed with Typhoon 9410 (GE Healthcare).

## Reporting summary

Further information on research design is available in the Nature Portfolio Reporting Summary linked to this article.

## Data availability

All data supporting the findings of this study are available within the main manuscript or Supplementary Materials. Biological materials (strains, plasmids, antibodies) are available from the corresponding authors on reasonable request. Motion corrected micrographs with annotated particle locations were deposited in EMPIAR with the accession number EMPIAR-12933[81]. The map and the model for Ssb-ADP S1 (complex of 80S with nascent FLAG-Pgk1-70 and Ssb1 in the S1 state) were deposited in the EMDB and PDB with the accession numbers 9R9O and EMD-53860. The map and the model for Ssb-ADP S2 (complex of 80S with nascent FLAG-Pgk1-70 and Ssb1 in the S2 state) can be found via 9R9P and EMD-53861. Source Data are provided as a Source Data file. Source data are provided with this paper.

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

## Acknowledgements

This work was supported by the Deutsche Forschungsgemeinschaft (DFG) through Project-ID 403222702 - SFB 1381, TP B08 (to S.R.) and TP B07 (to T.H.), RO 1028/5-2 (to S.R.), through Germany's Excellence Strategy BIOSS-2 and CIBSS (to S.R and T.H.), and Boehringer Ingelheim (to D.H. and L.G.). Cryo-EM sample preparation and screening was performed in the Electron Microscopy Facility at Vienna BioCenter Core Facilities (VBCF), member of the Vienna BioCenter (VBC), Austria.

## Author contributions

Y.Z., L.G., L.V., J.S., V.H., M.A., I.G., A.M., and K.W., designed and performed experiments. Y.Z., L.G., T.H, D.H., and S.R. designed experiments and analyzed the data, S.R. wrote the manuscript. All authors commented on the manuscript.

## Funding

## Competing interests

The authors declare no competing interests.
