## [Transparent Peer Review file · Nature Communications]

The cotranslational cycle of the ribosome-bound Hsp70 homolog Ssb

Corresponding Author: Professor Sabine Rospert

Editorial Note: Parts of this Peer Review File have been redacted as indicated to remove third party material where no permission to publish were obtained.

Version 0:

Reviewer comments:

Reviewer #1

(Remarks to the Author)

The manuscript by Zhang, Grundmann et al. presents cryo-EM structures of the yeast Hsp70 homolog Ssb bound to the ribosome. This fills a distinct knowledge gap in the field regarding how this protein folding chaperone can cotranslationally engage with substrates. The single particle cryo-EM work is well done, and the interactions visualized in the structures are nicely validated with biochemical assays and careful mutant analyses. This work also convincingly demonstrates an impact of nascent protein engagement on the salt-resistant binding stability of Ssb to ribosomes. Overall, these results reflect careful, high quality experimentation. However, I found the final mechanistic model to be confusing and unnecessarily complicated. I believe the manuscript would be improved if the authors did not attempt to seemingly incorporate every single observation related to RAC and Ssb into a comprehensive model and instead focused on emphasizing functional concepts that are clearly demonstrated versus those that may be consistent with alternative interpretations.

Specific comments:

1. Figure 5 presents a mechanistic model with incredibly detailed but incompletely supported steps, in an apparent attempt to explain multiple conformational states of RAC bound to the ribosome (4 states), and separately, of Ssb bound to the ribosome (2 states). Although I appreciate the effort put into reconciling all these states, it does not seem necessary, and many of the steps appear to be conjecture. In my opinion, these rather confusing and detailed analyses detracts from the high-quality work in this study. I would favor removing most of these descriptions and significantly simplifying the Discussion for several reasons:

- a. The RAC conformational states were reported in previous studies, and there is not enough space to clearly explain what the four different states are. As a result, the Discussion covering these states is incredibly dense with overly specialized nomenclature.
 - b. It is not clear to me why all states of RAC and Ssb observed by cryo-EM must be considered functionally relevant or part of the same mechanistic pathway. There are numerous examples of parallel factor-ribosome engagement pathways. Cryo-EM also may capture states that are artificially stabilized during lysis and isolation procedures. It does not seem like there is sufficient evidence to suggest that all of the observed states must be put into the same pathway, that they are as precisely coordinated as indicated in Figure 5, or that Ssb loading may not occur slightly differently for different ribosome-nascent protein complexes.
2. The nascent protein density outside of the ribosomal exit tunnel is not clear in the provided maps. Is it only visible in unsharpened maps? The maps provided also appear to have gone through a sharpening procedure, but sharpening B-factors are not provided in the statistics table.
3. How was ADP assigned to the ribosome-bound Ssb in the structure? Was this by inference or by structural modeling? This reasoning could be clearer.

Reviewer #2

(Remarks to the Author)

In the manuscript "The cotranslational cycle of the ribosome-bound Hsp70 homolog Ssb", Ying Zhang and coworkers present a structure of the Hsp70 chaperone, Ssb, bound to the yeast ribosome. They observe good density for the SBDA domain, which allows them to localise a contact with Rpl25. This interaction interface is then tested with a series of experiments involving mutagenesis and binding assays. Finally, the authors combine this new structure with the previous structure of RAC to provide a theoretical model for the cotranslational protein biogenesis process. The structure of ribosome-

bound Ssb is a significant result as it is one of the best (higher resolution) structures reported to date, to my knowledge, for this important system.

From the cryoEM perspective, the presented structures support the main conclusion regarding the interaction interface. Specifically, the cryoEM map in this region is of good quality, allowing the authors to place SBDa confidently and to describe the potential interaction interface between the SBDa domain and Rpl25. Namely, a complementary electrostatic patch is identified on Ssb and Rpl25. The mutagenesis and binding experiments that validate this potential electrostatic interaction appear to be well-performed and logical.

Specific Comments:

- 1) In the main text, the nature of the studied complex is not well explained and has to be understood from the Methods section. For example, it could be clearer that the complex is prepared in vitro by mixing purified protein with RNCs made from an in vitro translation reaction (as opposed to isolating an in vivo assembled complex). The methods also mention that a protein, His6-Snl1- Δ TM, was also added, but it is not well explained what its role is, and if it is observed in the structure.
- 2) It would be useful to make it clearer to the reader exactly which parts of Ssb's structure were solved in relation to the ribosome. This could be facilitated by including the domain structure drawing of Ssb (Extended Data Figure 1B) as part of Figure 1 and also indicating in the text which residues are truly solved at high resolution, which are primarily docked from a model and which are disordered. In line with this, a close-up view of the SSB density (i.e., using a volume zone in ChimeraX or a segmented map), coloured according to local resolution, with the model inside, would help convey the confidence in the SBDa model, which forms the binding interface. Similarly, the lower local resolution in SBDb could indicate inherent flexibility in SBDb, which could be important for function.
- 3) The text indicates that Ssb is in the ADP form, but it is unclear if ADP or ATP is in the buffer or co-purifies with the protein. I guess the ADP form is inferred from the domain arrangement of Ssb, but this could be clarified in the text.
- 4) Neither of the figures shows density for all 17 aa of the NC. A more complete figure would help the reader understand if one can assign sidechains for all 17 (line 64) or only bulky residues (i.e. Tyr).
- 5) In Figure 2A, the nature of the map shown is unclear. Is it unsharpened, sharpened or low-pass filtered? The sharpened map provided appears to show more details than this figure. In the main text, it could be made more explicit that Figure 2A refers to State S1 and the extended data Figure 5A to State S2, as the figures are very similar on first glance. The interaction could be described better as some of the mentioned residues are positioned to interact directly, while others are more indirect.
- 6) Line 653: It could be useful to explain how the mask was generated.
- 7) Line 644: More details could be provided for the training, for example, the encoder/decoder dimensions and number of layers, number of epochs, and whether filtering was done. In the cryodrgon step, a lot of particle projections are eliminated. Are these the rotated ribosomes? Does the rotation state have a substantial effect on Ssb binding, and if so, why would this be? Would it be possible to combine rotated and non-rotated particles by using a refinement focused on the large subunit (or maybe using particle subtraction, multibody refinement)? A larger dataset could improve the density. Has this been tried to no effect?
- 8) Figure 1D The rendering of the density in the figure does not really give it a 3D feel to judge if density for the NC is present or not. Moreover, the low local resolution of SBDb domain, in general, could make it challenging to say the NC is there confidently (i.e. line 63-64). That said, there is an extension (colored in red in Figure 1D) that leads out of the SBDb domain towards the tunnel exit. Perhaps the text "clear nascent chain density in both states" could be rephrased as follows: "There is an extension (red density, Figure 1D) leading away from the SBDb density towards the NC tunnel exit, which we tentatively attribute to the NC."
- 9) In the local resolution panels, does red indicate 6.9 Å or does it indicate > 6.9 Å? I mean, for example, when colouring the map, is the range set to a max of 6.9, so any voxels with a value of 6.9 or above are rendered in red? What package was used for the local resolution calculation?

Minor Comments:

- 1) The company Thermo Fisher Scientific is referred to as both "ThermoFischer Scientific" and "Thermo Scientific" in the methods. It should be consistent and spelt correctly
- 2) Line 632, the provided magnification is not consistent with Table 2
- 3) Table 1 indicates the FSC threshold is 0.5, but in the Extended Data Figure 2B and E, it appears that 0.143 is used
- 4) Line 807: In "EMPIAR-XYZ", XYZ can be updated.
- 5) Extended Data Fig. 2. The number of particles in the lower part of the State S1 side should be double-checked. The figure

indicates that 1.28×10^5 particles go into the 3D classification, then 53% are selected. In the next step after the selection, the particles have increased to 1.31×10^5 and again increase after the next classification to 6.14×10^5 .

6) Extended Data Figure 5 in some binding assays, the fractions are designated T, C, and R and in other T, S, and R (i.e. Extended Data Figure 7). The reason for the difference is unclear.

7) Extended Data Figure 5D. Is it possible the western blots have been shifted up, as here Ssb and rpl4 run at 116 and 66 kDa, while in Extended Data Figure 5G they run at 66 and 35 kDa

8) Extended Data Figure 4 right-hand panels. It would be useful to have the L25 label.

Reviewer #3

(Remarks to the Author)

Co-translational folding of proteins by chaperones is a critical process in protein homeostasis. Hsp70s are an abundant class of chaperones that interact with and facilitate folding of nascent chains. In yeast, the Hsp70 Ssb has a cochaperone, RAC, comprising Zuo1 and Ssz1. How Ssb and RAC function together on the ribosome is not well understood. In this manuscript, the authors use single particle cryo-EM as well as biochemical and genetic approaches to examine this question. The principal finding of the manuscript is a high-resolution structure of Ssb bound to programmed ribosome nascent chain complexes. The authors find that the primary binding site for Ssb1 is the ribosomal protein Rpl25 (uL23). uL23 is at the exit tunnel and this binding site is consistent with previous crosslinking data and by convincing genetic and biochemical experiments presented in this manuscript. The authors attempt to address the question of how Ssb and RAC function is coordinated by using AlphaFold predictions and proposing a complex series of domain rearrangements. However, this is highly speculative and largely leaves open the issue of the coordination between Ssb1 and RAC. Overall, the experiments are cleanly executed and presented, however the manuscript would have greater impact if more concrete experimental evidence were presented for the interplay between Ssb and RAC.

Major points

1. Figure 1 Can the authors comment on what contributes to the differential positioning of SBD- β in the S1 and S2 structures? Does it make specific contacts with the ribosome or are these different conformations intrinsic to Ssb itself?

2. In Figure 3f, the salient question is whether or not the residual ribosome binding activity of Ssb-DD1-DD2 is due to Ssz1. The synthetic growth defect is suggestive but indirect. The authors should directly test if Ssz1 is responsible for the residual Ssb binding by measuring binding of Ssb1-3*-DD1-DD2 to ribosomes or the binding of Ssb-DD1-DD2 to ribosomes in the absence of Ssz1. To support the genetic interaction, the authors should also show that this protein expresses well, so that the binding and growth defects cannot be attributed to poor expression of the mutant protein.

3. In Figure 5, the authors explain that Ssb and Rac initially bind simultaneously, followed by a hand off to Ssb. They also argue that Ssz1 provides a binding site for Ssb1, see figure 3f. However, the data in figure 4d argue that the two complexes are not present on the ribosome at the same time. These results seems inconsistent. The authors generated a nice set of truncated PGK1 vectors that allows the production of RNCs of different lengths. Can they show that RAC does bind on short nascent chains and that Ssb predominates as chains grow in length?

Minor points

4. Panel D of Figure 2 is conceptually unrelated to the rest of the figure and should be moved to supplementary data. Furthermore, the idea being tested here is that mutations in Rpl25 affect SRP binding. However, the readout is quite indirect - whether or not expression of a substrate for SRP is affected. Directly examining SRP binding to Rpl25-KKK ribosomes would be more direct.

5. Figure 3 introduces the DD1-DD2 mutant. Including a cartoon of this region of Ssb and the substitutions in the figure would help the reader understand exactly what the mutation is without having to refer to the supplemental materials.

6. Figure 4a Because of the scale of the x-axis, the curves are not well separated. Including an inset for ribosome concentration range $<1 \mu\text{M}$ would be helpful.

7. L44 Please explain the composition of RAC here rather than referring the reader to the supplemental materials.

8. Please explain how ADP was assigned in the cryo-EM structures if the nucleotide binding domain was not resolved.

9. In figure 5, the cartoons are too small to convey the conformational changes being proposed.

Version 1:

Reviewer comments:

Reviewer #1

(Remarks to the Author)

The revised manuscript addresses my comments and I support publication.

Reviewer #2

(Remarks to the Author)

I would like to thank the authors for their thorough response and revision. The manuscript has improved in both clarity and rigour. From my side, the responses to the review comments are detailed and convincing, and I find that my previous concerns have been largely addressed. I have no further major comments.

A minor question/clarification would be how the rotation axis in Supplementary Fig. 3D was determined. Also in Figure 3 are the volumes shown from a backprojection (cryodrgn backproject_voxel) of all particles in the class, or the volume at the centre of the class?

Reviewer #3

(Remarks to the Author)

REPLY TO REVIEWER COMMENTS

We are pleased that all three reviewers were positive regarding our manuscript. Moreover, we wish to thank each reviewer for their insightful and helpful suggestions. We feel that by implementing the suggested changes, the manuscript has been significantly improved. Our replies are in blue. When appropriate, we have copied and pasted text passages (low res) from the revised manuscript or Supplementary Data for the Reviewers' convenience.

Reviewer #1 (Remarks to the Author):

The manuscript by Zhang, Grundmann et al. presents cryo-EM structures of the yeast Hsp70 homolog Ssb bound to the ribosome. This fills a distinct knowledge gap in the field regarding how this protein folding chaperone can cotranslationally engage with substrates. The single particle cryo-EM work is well done, and the interactions visualized in the structures are nicely validated with biochemical assays and careful mutant analyses. This work also convincingly demonstrates an impact of nascent protein engagement on the salt-resistant binding stability of Ssb to ribosomes. Overall, these results reflect careful, high quality experimentation. However, I found the final mechanistic model to be confusing and unnecessarily complicated. I believe the manuscript would be improved if the authors did not attempt to seemingly incorporate every single observation related to RAC and Ssb into a comprehensive model and instead focused on emphasizing functional concepts that are clearly demonstrated versus those that may be consistent with alternative interpretations.

Specific comments:

1. Figure 5 presents a mechanistic model with incredibly detailed but incompletely supported steps, in an apparent attempt to explain multiple conformational states of RAC bound to the ribosome (4 states), and separately, of Ssb bound to the ribosome (2 states). Although I appreciate the effort put into reconciling all these states, it does not seem necessary, and many of the steps appear to be conjecture. In my opinion, these rather confusing and detailed analyses detracts from the high-quality work in this study. I would favor removing most of these descriptions and significantly simplifying the Discussion for several reasons:

a. The RAC conformational states were reported in previous studies, and there is not enough space to clearly explain what the four different states are. As a result, the Discussion covering these states is incredibly dense with overly specialized nomenclature.

We agree with the above comments of Reviewer #1. In order to simplify the model and focus on the central observations of this work we made the following changes:

- We removed all hybrid RAC models (RAC S1, RAC S2, RAC S3, and RAC S4) from the model depicted in Fig. 5.

- We now present a simplified model that involves (i) the three Ssb states that we characterize in this work: Ssb-ADP S1, Ssb-ADP S2 (cryo-EM work of this study), and Ssb-ATP (model based on the known structure of Ssb-ATP ¹ and biochemical data presented in this manuscript (Fig. 3e,f and Fig. 4a), (ii) the atomic model of RAC C2 (PDB 7X3K, ²), and (iii) an AlphaFold 3 model generated with the input Ssb, Zuo1, Ssz1, which we validated carefully (Supplementary Note 2 and Supplementary Fig. 10)

- We have significantly simplified the model presented in Fig. 5 as well as the Discussion by reducing the intermediates and by refraining from integrating all published data as suggested by Reviewer #1. This allowed us to focus more clearly on the essential results of this work. We feel that these changes improved the value of the manuscript.

b. It is not clear to me why all states of RAC and Ssb observed by cryo-EM must be considered functionally relevant or part of the same mechanistic pathway. There are numerous examples of parallel factor-ribosome engagement pathways. Cryo-EM also may capture states that are artificially stabilized during lysis and isolation procedures. It does not seem like there is sufficient evidence to suggest that all of the observed states must be put into the same pathway, that they are as precisely coordinated as indicated in Figure 5, or that Ssb loading may not occur slightly differently for different ribosome-nascent protein complexes.

We agree with Reviewer #1 and no longer consider all cryo-EM structures (see our reply to comment a) above). We have revised the model depicted in Fig. 5 and adjusted the Discussion accordingly. We also clarify in the Discussion that alternative pathways of the RAC-Ssb system may exist.

~~3e f) acts cooperatively with binding of Ssb-ATP to Ssz1 (Fig. 3g h). On the basis of our data, and in consideration of previous observations, we propose a model (Fig. 5) that provides a consistent explanation for how the RAC-Ssb system fulfills the spatial and temporal constraints described above. We would like to note, however, that parallel pathways may exist, and that the properties of the nascent chain may influence the sequence of events during the RAC-Ssb cycle.~~

2. The nascent protein density outside of the ribosomal exit tunnel is not clear in the provided maps. Is it only visible in unsharpened maps? The maps provided also appear to have gone through a sharpening procedure, but sharpening B-factors are not provided in the statistics table.

The nascent chain density outside of the tunnel as shown in Fig. 1d upon submission was changed according to the suggestions of Reviewer #2. Fig. 1d now shows cross sections through lowpass filtered (sdev=2.5), non-post-processed cryo-EM maps of Ssb-ADP S1 and S2. The previous Fig. 1d is now shown as Supplementary Figure 1f.

The legend of Fig. 1d (*) and the Results section (**) now contain a more detailed description of the visualization of nascent chain density. Both the Ssb-ADP S1 and S2 non-post-processed maps, as well as the corresponding post-processed maps, are available now as EMD-53860 and EMD-53861. We apologize for not mentioning the deposition of the maps in our original submission. In addition, the unsharpened maps have been uploaded to the figshare repository.

* Figure 1d

~~mRNA base pairing at the peptidyl transferase center. (d) Binding of the nascent chain to the peptide-binding cleft of Ssb-SBD β . Cross-section through lowpass filtered (sdev=2.5), non-post-processed maps of Ssb-ADP S1 and S2. (e) Close-up of the contacts of Ssb-ADP S1 with~~

** Results

of SBD β positioned close to the tunnel exit (Fig. 1e and Supplementary Fig. 5a,b). The inherent flexibility of nascent FLAG-Ppk1-70 bound to Ssb-SBD β prevented direct visualization at high resolution, however, low-pass filtered non-sharpened maps revealed an extension leading away from the ribosomal tunnel exit to the SBD β substrate-binding cleft, which we tentatively attributed to the nascent chain (Fig. 1d and Supplementary Fig. 1f, red density). ¶

Supplementary Fig. 1f (previous Fig. 1d)

nascent chain. (f) Binding of the nascent chain to the peptide-binding cleft of Ssb-SBD β . Overlays of cross-sections through gaussian filtered (sdev=2.5), non-post-processed cryo-EM maps and corresponding atomic models of Ssb-ADP S1 and S2. The nascent chain model was placed by aligning PDB 1DKZ⁴ into Ssb-SBD β of Ssb-ADP S1 and S2. ¶

With regard to the sharpening B-factor, we would like to clarify that the post-processed maps were not sharpened in the classical sense. Instead, we used the tool deepEMhancer³, an automated deep-learning-based post-processing algorithm that does not rely on B-factor estimation. Consequently, no B-factor was applied during post-processing and, therefore, is not provided in the statistics table. Details can be found in Methods (***) .

See also our reply to Reviewer #2 and Fig. 1d and Supplementary Fig. 1f.

*** Methods:

Particles with more defined tRNA density were selected for homogeneous refinement. The resulting maps were post-processed via DeepEMhancer using the 'high-res' weights provided by the developers⁶³ resulting in cryo-EM maps Ssb-ADP S1 and S2 (Fig. 1b). Local resolution estimation was performed in CryoSPARC⁵⁸. ¶

3. How was ADP assigned to the ribosome-bound Ssb in the structure? Was this by inference or by structural modeling? This reasoning could be clearer.

We apologize for having omitted this essential information and thank the Reviewers for drawing our attention to this important point. As it was raised by all three Reviewers, our response is addressed to Reviewers #1-3.

Reviewer 1: 3. How was ADP assigned to the ribosome-bound Ssb in the structure? Was this by inference or by structural modeling? This reasoning could be clearer.

Reviewer 2: 3) The text indicates that Ssb is in the ADP form, but it is unclear if ADP or ATP is in the buffer or co-purifies with the protein. I guess the ADP form is inferred from the domain arrangement of Ssb, but this could be clarified in the text.

Reviewer 3: 8. Please explain how ADP was assigned in the cryo-EM structures if the nucleotide binding domain was not resolved.

The nucleotide binding domain (NBD) of Ssb was not resolved in the Ssb-ADP S1/S2 structures; therefore, the nucleotide bound to the Ssb-NBD could not be assessed directly. Instead, we inferred the nucleotide state from the conformation of the substrate-binding domain (SBD) in the cryo-EM maps (Fig. 1b). In both Ssb-ADP structures, the conformation of the Ssb-SBD closely resembled that of the ADP-bound DnaK-SBD (PDB 2KHO⁴). Specifically, Ssb-SBD α (the "lid domain") was observed in a closed position over the Ssb-SBD β substrate-binding cleft. By contrast, the SBD of ATP-bound Ssb (and

Hsp70s in general) adopts a distinct open conformation (PDB 5TKY⁵), which was incompatible with the conformation of ribosome-bound Ssb observed in our cryo-EM maps.

We have improved the Introduction by providing a more detailed description of the distinct conformations of Hsp70 proteins and their functional significance. In the Results section, we now explicitly state that the conformation of the Ssb-SBD is consistent exclusively with the ADP-bound state.

Introduction:

(Supplementary Fig. 1a). Yeast Ssb is a canonical Hsp70 (encoded by the nearly identical *SSB1* and *SSB2* genes), consisting of an N-terminal nucleotide-binding domain (NBD) and a C-terminal substrate-binding domain (SBD) (Fig. 1a). The NBD possesses ATPase activity, the SBD is subdivided into a β -sheet domain (SBD β) and an α -helical lid domain (SBD α) (Fig. 1a). As in other canonical Hsp70s, the two domains of Ssb are connected via a linker that allows allosteric coupling of ATP hydrolysis with tight substrate binding to the SBD (Supplementary Fig. 1a). When Ssb is bound to ATP, the linker interacts with the NBD, and the SBD α lid adopts an open conformation⁵. Upon ATP hydrolysis, significant structural rearrangements occur, which lead to the detachment of the linker from the NBD, allowing the SBD α lid to close over the substrate-binding pocket on SBD β , thereby stabilizing substrate binding.^{1,4,6}¶

Results:

Cryo-EM structure of ADP-bound Ssb associated with translating ribosomes. To determine the structure of Ssb associated with translating ribosomes, ribosome-nascent chain complexes (RNCs) carrying FLAG-tagged 3-phosphoglycerate kinase (FLAG-Pgk1-70; Supplementary Fig. 1c-f) were generated in a yeast *in vitro* translation system in the presence of Ssb, were subsequently purified by FLAG affinity chromatography, and were analyzed by cryo-EM (Supplementary Fig. 2; see Methods). In these samples, ribosomes representing diverse states along the translational cycle were detected. Notably, non-rotated RNCs containing a P-site tRNA and associated with the density of a trailing ribosome within a polysome displayed enhanced density definition near the ribosomal tunnel exit (Supplementary Fig. S3). Further analysis of these particles (Supplementary Fig. S2) yielded two cryo-EM structures in which Ssb was bound to ribosomes stalled in a non-rotated, post-translocation conformation with a P-site Met-elongator tRNA (tRNA^{Met-e}). In both structures, the SBD of Ssb (Fig. 1b and Supplementary Table 1) was in the closed conformation characteristic for the ADP-bound state of Hsp70s, in which SBD α is tightly packed against SBD β (see Introduction). The Ssb-NBD was not visible in the cryo-EM maps, consistent with the known flexibility between the NBD and SBD in ADP-bound Hsp70s.^{4,29,30} The resolution of the complexes,

Reviewer #2 (Remarks to the Author):

In the manuscript "The cotranslational cycle of the ribosome-bound Hsp70 homolog Ssb", Ying Zhang and coworkers present a structure of the Hsp70 chaperone, Ssb, bound to the yeast ribosome. They observe good density for the SBDa domain, which allows them to localise a contact with Rpl25. This interaction interface is then tested with a series of experiments involving mutagenesis and binding assays. Finally, the authors combine this new structure with the previous structure of RAC to provide a theoretical model for the cotranslational protein biogenesis process. The structure of ribosome-bound Ssb is a significant result as it is one of the best (higher resolution) structures reported to date, to my knowledge, for this important system.

From the cryoEM perspective, the presented structures support the main conclusion regarding the interaction interface. Specifically, the cryoEM map in this region is of good quality, allowing the authors to place SBDa confidently and to describe the potential interaction interface between the SBDa domain and Rpl25. Namely, a complementary electrostatic patch is identified on Ssb and Rpl25. The mutagenesis and binding experiments that validate this potential electrostatic interaction appear to be well-performed and logical.

Specific Comments:

1) In the main text, the nature of the studied complex is not well explained and has to be understood from the Methods section. For example, it could be clearer that the complex is prepared *in vitro* by mixing purified protein with RNCs made from an *in vitro* translation reaction (as opposed to isolating an *in vivo* assembled complex). The methods also mention that a protein, His6-Snl1- Δ TM, was also added, but it is not well explained what its role is, and if it is observed in the structure.

Description of sample preparation:

We thank Referee #2 for pointing that out that we need to detail the nature of Ssb-ribosome complexes employed for cryo-EM in the Results. Please note that Ssb was not added to purified RNCs, but was present during the translation reaction.

Results:

~~Cryo-EM structure of ADP-bound Ssb associated with translating ribosomes. To determine the structure of Ssb associated with translating ribosomes, ribosome-nascent chain complexes (RNCs) carrying FLAG-tagged 3-phosphoglycerate kinase (FLAG-Pgk1-70; Supplementary Fig. 1c-f) were generated in a yeast *in vitro* translation system in the presence of Ssb, were subsequently purified by FLAG affinity chromatography, and were analyzed by cryo-EM (Supplementary Fig. 2, see Methods). In these samples, ribosomes representing~~

We now also detail the components of the translation reactions in Methods. To drive translation, the translation buffer contained ATP, GTP, and creatine phosphate/creatine phosphate kinase. During subsequent FLAG-affinity purification of RNCs nucleotide was omitted from the buffers.

Methods:

indicated in the Figure Legends.^{21,53} Translation reactions were performed with 35% yeast translation extract in translation buffer (1.0 mM ATP, 0.08 mM GTP, 17.5 mM creatine phosphate, 0.2 mg/ml creatine phosphokinase, 0.2 mM of each amino acid, 2.2 mM putrescine, 0.1 mg/ml yeast tRNA, 100 U/ml Ribolock (Thermo-Fisher-Scientific)) at 20°C as previously described.^{53,54} After 80 min incubation at 20°C translation reactions were stopped by addition

Role of His₆-Snl1-ΔTM

We thank the Reviewer for pointing out that we had not explained the rationale for including Snl1-ΔTM during *in vitro* translation. The rationale was that the BAG-domain protein Snl1-ΔTM, lacking its N-terminal transmembrane domain, was suggested to facilitate the recruitment of Ssb to translating ribosomes: "*We postulate that Snl1 represents an additional method by which fungi recruit Hsp70 to translating ribosomes to support protein biogenesis.*"⁶.

For this reason, we carried out *in vitro* translations with or without Snl1-ΔTM, isolated RNCs via the FLAG-tagged nascent chain, prepared cryo-EM grids, and collected single-particle analysis datasets for both samples. After applying comparable particle-cleaning procedures (Fig. R1), we observed that the reconstruction derived from the sample containing Snl1-ΔTM exhibited more defined density near the polypeptide exit tunnel compared to the reconstruction without Snl1-ΔTM (Fig. R1). Importantly, however, we were unable to detect any density corresponding to Snl1-ΔTM in the cryo-EM density maps. These data (Fig. R1) suggest a positive role for Snl1-ΔTM in facilitating Ssb recruitment to RNCs, possibly by enhancing either the recruitment process itself or promoting uniform positioning of Ssb at the tunnel exit. Nevertheless, because this inference is based on a single dataset and replicate, the available evidence is insufficient to draw a definitive conclusion. Differences in sample preparation and/or grid handling, or discrepancies during data collection offer alternative - and more likely - explanations for the variation observed in both reconstructions. In the interest of scientific rigor, we therefore decided to avoid overemphasizing this preliminary observation in the manuscript, while taking advantage of the better quality of the Snl1-ΔTM dataset for further analysis. However, we now provide more detailed information on Snl1-ΔTM in the Methods Section.

Figure R1. The dataset from the *in vitro* translation reaction supplemented with Sn11-ΔTM shows a more defined density for Ssb1. The processing scheme outlines the steps used to obtain the initial 80S ribosome consensus reconstruction, which served as the basis for downstream processing of RNCs purified from *in vitro* translation reactions supplemented with (a) Ssb and Sn11-ΔTM, and (b) Ssb alone. Shown are gold standard Fourier shell correlation plots (GSFSC plots) used for global resolution estimation, along with viewing angle distribution plots. Front and top views of the unsharpened (non-post-processed) maps are presented. Insets highlight the Ssb binding site, with Gaussian-filtered maps (sdev = 2.5) shown transparently, the unsharpened maps in gray, and the Ssb1-ADP S1 model in salmon. Thresholding values used for the filtered maps in ChimeraX (v.1.9) are indicated at the bottom.

Methods:

Purification of RNCs and preparation of cryo-EM grids. RNCs carrying FLAG-Ppk1-70 were generated in Δ ssb1 Δ ssb2 extract complemented with purified, untagged Ssb1, or complemented with untagged Ssb and His-Sn11-ΔTM (Supplementary Data-Fig. 6f) each at a final concentration of 3 μ M. His-Sn11-ΔTM was included in the translation reaction because it was previously suggested that the BAG-domain protein Sn11, lacking its single transmembrane domain (Sn11-ΔTM) facilitates the recruitment of Ssb to translating ribosomes⁵⁶. For the

Data collection and image processing. Cryo-EM data were collected on grids with FLAG-Pgk1-70 RNCs prepared in the presence of Ssb, with or without His₆-Snl1-ΔTM. Initial reconstructions revealed Ssb density near the tunnel exit in both samples, while no density for His₆-Snl1-ΔTM was observed. We observed that the sample containing His₆-Snl1-ΔTM showed more clearly defined Ssb density, which could potentially indicate a positive effect of Snl1-ΔTM on Ssb recruitment or positioning at the tunnel exit. However, we consider it more likely that this variation reflects the intrinsic variability of cryo-EM data obtained from affinity purified, *in vitro*-generated RNCs, as well as differences in cryo-EM sample preparation, grid handling, and data collection. As more defined Ssb density provided an advantage for subsequent data collection, the His₆-Snl1-ΔTM dataset was selected for further analysis. Movies were collected

2) It would be useful to make it clearer to the reader exactly which parts of Ssb's structure were solved in relation to the ribosome. This could be facilitated by including the domain structure drawing of Ssb (Extended Data Figure 1B) as part of Figure 1 and also indicating in the text which residues are truly solved at high resolution, which are primarily docked from a model and which are disordered. In line with this, a close-up view of the SSB density (i.e., using a volume zone in ChimeraX or a segmented map), coloured according to local resolution, with the model inside, would help convey the confidence in the SBD α model, which forms the binding interface. Similarly, the lower local resolution in SBD β could indicate inherent flexibility in SBD β , which could be important for function.

- We now show the Ssb domain structure of Ssb in Fig. 1a.

- We now show the Ssb density colored according to local resolution, with the molecular model inside, in the new Supplementary Figures 4d and 4h. These Figures reveal that helix Ssb- α D, which forms the interface with Rpl25, exhibits the highest resolution within Ssb-SBD.

- we now state more clearly which parts of the Ssb-SBD were solved at high resolution, which were primarily docked from a model, and which are disordered.

Supplementary Fig. 1f. Local resolution of Ssb-SBD enabled side chain assignment for SBD α . For SBD β , flexibility relative to SBD α and the translating ribosome hindered side chain assignment; therefore, the SBD β backbone was modeled as a rigid body (Supplementary Fig. 4d,h). Ssb-ADP was attached to the ribosome mainly via SBD α with the peptide-binding cleft of SBD β positioned close to the tunnel exit (Fig. 1e and Supplementary Fig. 5a,b). The inherent flexibility of nascent FLAG-Pgk1-70 bound to Ssb-SBD β prevented direct visualization at high resolution, however, low-pass filtered non-sharpened maps revealed an extension leading away from the ribosomal tunnel exit to the SBD β substrate binding cleft, which we tentatively attributed to the nascent chain (Fig. 1d and Supplementary Fig. 1f, red density). ¶

3) The text indicates that Ssb is in the ADP form, but it is unclear if ADP or ATP is in the buffer or co-purifies with the protein. I guess the ADP form is inferred from the domain arrangement of Ssb, but this could be clarified in the text.

We apologize for having omitted this essential information and thank the Reviewers for drawing our attention to this important point. As it was raised by all three Reviewers, our response is addressed to Reviewers #1-3.

Reviewer 1: 3. How was ADP assigned to the ribosome-bound Ssb in the structure? Was this by inference or by structural modeling? This reasoning could be clearer.

Reviewer 2: 3) The text indicates that Ssb is in the ADP form, but it is unclear if ADP or ATP is in the buffer or co-purifies with the protein. I guess the ADP form is inferred from the domain arrangement of Ssb, but this could be clarified in the text.

Reviewer 3: 8. Please explain how ADP was assigned in the cryo-EM structures if the nucleotide binding domain was not resolved.

The nucleotide binding domain (NBD) of Ssb was not resolved in the Ssb-ADP S1/S2 structures; therefore, the nucleotide bound to the Ssb-NBD could not be assessed directly. Instead, we inferred the nucleotide state from the conformation of the substrate-binding domain (SBD) in the cryo-EM maps (Fig. 1b). In both Ssb-ADP structures, the conformation of the Ssb-SBD closely resembled that of the ADP-bound DnaK-SBD (PDB 2KHO⁴). Specifically, Ssb-SBD α (the “lid domain”) was observed in a closed position over the Ssb-SBD β substrate-binding cleft. By contrast, the SBD of ATP-bound Ssb (and Hsp70s in general) adopts a distinct open conformation (PDB 5TKY⁵), which was incompatible with the conformation of ribosome-bound Ssb observed in our cryo-EM maps.

We have improved the Introduction by providing a more detailed description of the distinct conformations of Hsp70 proteins and their functional significance. In the Results section, we now explicitly state that the conformation of the Ssb-SBD is consistent exclusively with the ADP-bound state.

Introduction:

(Supplementary Fig. 1a). Yeast Ssb is a canonical Hsp70 (encoded by the nearly identical *SSB1* and *SSB2* genes), consisting of an N-terminal nucleotide-binding domain (NBD) and a C-terminal substrate-binding domain (SBD) (Fig. 1a). The NBD possesses ATPase activity, the SBD is subdivided into a β -sheet domain (SBD β) and an α -helical lid domain (SBD α) (Fig. 1a). As in other canonical Hsp70s, the two domains of Ssb are connected via a linker that allows allosteric coupling of ATP hydrolysis with tight substrate binding to the SBD (Supplementary Fig. 1a). When Ssb is bound to ATP, the linker interacts with the NBD, and the SBD α lid adopts an open conformation⁵. Upon ATP hydrolysis, significant structural rearrangements occur, which lead to the detachment of the linker from the NBD, allowing the SBD α lid to close over the substrate-binding pocket on SBD β , thereby stabilizing substrate binding.^{1,4,6}¶

Results:

Cryo-EM structure of ADP-bound Ssb associated with translating ribosomes. To determine the structure of Ssb associated with translating ribosomes, ribosome-nascent chain complexes (RNCs) carrying FLAG-tagged 3-phosphoglycerate kinase (FLAG-Pgk1-70; Supplementary Fig. 1c-f) were generated in a yeast *in vitro* translation system in the presence of Ssb, were subsequently purified by FLAG affinity chromatography, and were analyzed by cryo-EM (Supplementary Fig. 2; see Methods). In these samples, ribosomes representing diverse states along the translational cycle were detected. Notably, non-rotated RNCs containing a P-site tRNA and associated with the density of a trailing ribosome within a polysome displayed enhanced density definition near the ribosomal tunnel exit (Supplementary Fig. S3). Further analysis of these particles (Supplementary Fig. S2) yielded two cryo-EM structures in which Ssb was bound to ribosomes stalled in a non-rotated, post-translocation conformation with a P-site Met-elongator tRNA ($tRNA^{Met-e}$). In both structures, the SBD of Ssb (Fig. 1b and Supplementary Table 1) was in the closed conformation characteristic for the ADP-bound state of Hsp70s, in which SBD α is tightly packed against SBD β (see Introduction). The Ssb-NBD was not visible in the cryo-EM maps, consistent with the known flexibility between the NBD and SBD in ADP-bound Hsp70s^{4,29,30}. The resolution of the complexes,

4) Neither of the figures shows density for all 17 aa of the NC. A more complete figure would help the reader understand if one can assign side chains for all 17 (line 64) or only bulky residues (i.e. Tyr).

We thank Reviewer #2 for this helpful suggestion. To reveal the side chain density of nascent FLAG-Pgk1-70 residues, we have now included an overlay of the Ssb-ADP S2 FLAG-Pgk1-70 density with the corresponding atomic model in **Supplementary Fig. 1e**. Since the 3'-end of the truncated mRNA encoding FLAG-Pgk1-70 (Supplementary Fig. 1c) is well resolved in the atomic model (Supplementary Fig. 1e) and we were able to assign most of the 17 amino acid side chains, the backbone of the nascent chain was traced confidently. However, as Reviewer #2 correctly pointed out, not all side chains could be assigned based on nascent chain density. Accordingly, we have revised the text to reflect this limitation.

codon (Fig. 1b,c, Supplementary Fig. 1e, Supplementary Fig. 4a-h). The density of nascent FLAG-Pgk1-70 was well defined within the proximal region of the exit tunnel, allowing for side chain assignment of a large fraction of the 17 C-terminal residues and enabling confident modeling of the nascent chain (Fig. 1c and Supplementary Fig. 1e). While the nascent chain

5) In Figure 2A, the nature of the map shown is unclear. Is it unsharpened, sharpened or low-pass filtered? The sharpened map provided appears to show more details than this figure. In the main text, it could be made more explicit that Figure 2A refers to State S1 and the extended data Figure 5A to State S2, as the figures are very similar on first glance. The interaction could be described better as some of the mentioned residues are positioned to interact directly, while others are more indirect.

1. The overview maps in Figure 2a and Extended Data Figure 5a (now Supplementary Fig. 6a) were initially created using the molmap command in ChimeraX. We have now replaced these with post-processed cryo-EM maps. Additionally, the zoom into the helix Ssb- α D/Rpl25 interface is shown from both front and back views, with the distances between residues involved in ionic interactions indicated.

2. We have improved the description of the Ssb-SBD/Rpl25 interfaces in the S1 and S2 states and clarified their implications more clearly in the Results section.

Ssb interacts with ribosomal protein Rpl25 at the ribosomal tunnel exit. Analyses of the Ssb-ADP-ribosome interface in the atomic models of states S1 and S2 revealed that Ssb primarily contacts Rpl25 through salt bridges and long-range ionic interactions (Fig. 2a and Supplementary Fig. 6a). Key contributions to the interface came from Ssb residues R596, K597, K603, and R604 (termed the Ssb-RKKR motif, Supplementary Fig. 5c) and Rpl25 residues E77, D131, D134 (termed the Rpl25-EDD motif) and the Rpl25 carboxyl terminus (Fig. 2a and Supplementary Fig. 6a). Specifically, salt bridges were observed between Ssb-R604 and Rpl25-D131, Ssb-R596 and Rpl25-D134, and between Ssb-K603 and the C-terminus of Rpl25 (I142). Long-range ionic interactions were formed between Ssb-K597 and Rpl25-D134, and Ssb-K608 and Rpl25-E77. Although the overall geometry of the binding interface did not differ substantially between S1 and S2, Ssb-SBD β was closer to the ribosomal surface in S2 than in S1 (Fig. 1f and Supplementary Fig. 5a,b). To validate the

We feel that Fig. 2a, Supplementary Fig. 6a (previous Supplementary 5a), and the corresponding Results section are much improved by these changes, and we thank the Reviewer for his/her helpful comments.

6) Line 653: It could be useful to explain how the mask was generated.

We have added the following text to the Methods section and provide a more detailed explanation of the subsequent masking steps.

(10 classes, T = 300) masking around the Ssb density. The ellipsoid mask was generated in ChimeraX (v 1.7) using the 'shape' command followed by the 'volume onesmask' command. Subsequently, the volume was binarized and dilated by a 'soft-edge' (2 px, 2.64 Å/px) in CryoSPARC⁵⁸. The particles from collection 1 + 2 were combined with particles from the

7) Line 644: More details could be provided for the training, for example, the encoder/decoder dimensions and number of layers, number of epochs, and whether filtering was done. In the cryodrgn step, a lot of particle projections are eliminated. Are these the rotated ribosomes? Does the rotation state have a substantial effect on Ssb binding, and if so, why would this be? Would it be possible to combine rotated and non-rotated particles by using a refinement focused on the large subunit (or maybe using particle subtraction, multibody refinement)? A larger dataset could improve the density. Has this been tried to no effect?

Missing parameters were added to the Methods section.

classifications using CryoSPARC (v 4.2.1 to 4.5.1). *Ab-initio* reconstruction was performed on the first collection to obtain 80S and 60S reference volumes that were used for particle sorting in heterogeneous refinements. 80S particles were refined and used for cryoDRGN variational autoencoder training (128 px, 3.71 Å/px, 3 layers for encoder and decoder each, for 50 epochs in an 8-dimensional latent space⁶⁰). Using the cryoDRGN 'landscape' tool, latent space encodings were clustered via kmeans ($k = 1000$) and resulting reconstructions were clustered via agglomerative clustering ($k = 20$) (<https://ez-lab.gitbook.io/cryodrgn/>). Latent space

In response to the reviewer's suggestion, we have now included a new Supplementary Fig. 3 to illustrate the structural variability of RNCs as revealed by cryoDRGN. Shown are the clustering results for Collection 1 (Supplementary Figure 2) representative for the two other collections. The analysis identified several distinct clusters. The atomic models based on Cluster 1 particles (Ssb-ADP S1 and S2, Fig. 1b) showed that the ribosome had translated up to the most 3'-proximal AUG codon of the truncated FLAG-Pgk1-70 mRNA (Fig. 1c and Supplementary Fig. 1c,e). The projections removed by clustering of the cryoDRGN encoded latent space are indeed rotated as well as non-rotated ribosomes along the translational cycle with varying occupancy of the EPA sites by tRNAs. Also, differences in the peripheral densities corresponding to the leading or trailing ribosomes in a polysome were identified. These analyses demonstrated that ribosomal state - defined by rotation, tRNA occupancy, and polysome context - substantially affects the definition of Ssb at the polypeptide exit tunnel. In particular, non-rotated RNCs carrying a P-site tRNA and accompanied by a trailing ribosome exhibited markedly increased Ssb density compared to other ribosomal configurations (see Supplementary Fig. 3). In summary, in our experimental setup, Cluster 1 ribosomes, which are stalled, non-rotated ribosomes⁷ associated with fully translated nascent FLAG-Pgk1-70 were the most likely candidates to be associated with Ssb-ADP (see above).

We now better define in the cryoDRGN step in Methods and Supplementary Fig. 3:

Methods:

in heterogeneous refinements. 80S particles were refined and used for cryoDRGN variational autoencoder training (128 px, 3.71 Å/px, 3 layers for encoder and decoder each, for 50 epochs in an 8-dimensional latent space⁶⁰). Using the cryoDRGN landscape tool, latent space encodings were clustered via kmeans ($k = 1000$) and resulting reconstructions were clustered via agglomerative clustering ($k = 20$) (<https://ez-lab.gitbook.io/cryodrgn/>). Latent space dimensionality reduction was performed by uniform manifold approximate and projection (UMAP). The resulting reconstructions of ribosomal states across the translation cycle were analyzed for density near the ribosomal tunnel exit, the hypothesized binding site of Ssb. Best resolved density was observed in the cluster representing non-rotated, post-translocation state ribosomes, with a P-site tRNA and a peripheral density corresponding to a trailing ribosome (Supplementary Fig. 3). Corresponding particles were reextracted (180 px boxsize,

The increased Ssb density on the stalled RNCs also explains why 3D classification of all particles from the initial consensus refinement using e.g. an ellipsoid mask enclosing the Ssb density, was without success in our hands. We hypothesize that the smaller fraction of Ssb bound particles in the consensus refinement precluded sufficient signal to noise ratio for successful maximum likelihood-based 3D classification in RELION. Using cryoDRGN to indirectly cluster for particles with increased Ssb binding, allowed downstream 3D classification in RELION.

Indeed, increasing the dataset size can enhance the quality of cryo-EM maps. To address this, we collected over 54,000 micrographs, which represents substantially more compared to the datasets used in most studies. We would like to note that improvements in cryo-EM map quality typically follow a logarithmic trend (as shown in res-log plots), indicating that achieving substantial further improvements would require an order of magnitude more micrographs, which is not practically feasible.

8) Figure 1D The rendering of the density in the figure does not really give it a 3D feel to judge if density for the NC is present or not. Moreover, the low local resolution of SBD β domain, in general, could make it challenging to say the NC is there confidently (i.e. line 63-64). That said, there is an extension (colored in red in Figure 1D) that leads out of the SBD β domain towards the tunnel exit. Perhaps the text "clear nascent chain density in both states" could be rephrased as follows: "There is an extension (red density, Figure 1D) leading away from the SBD β density towards the NC tunnel exit, which we tentatively attribute to the NC."

We changed Fig. 1d according to the suggestions of Reviewer #2. Fig. 1d now shows a cross sections through lowpass filtered (sdev=2.5), non-post-processed cryo-EM maps of Ssb-ADP S1 and S2. The previous Fig. 1d is now presented as Supplementary Fig. 1f.

We followed the suggestion of Reviewer #2 and changed the text in the Results accordingly:

~~of SBD β positioned close to the tunnel exit (Fig. 1e and Supplementary Fig. 5a,b). The inherent flexibility of nascent FLAG-Ppk1-70 bound to Ssb-SBD β prevented direct visualization at high resolution, however, low-pass filtered non-sharpened maps revealed an extension leading away from the ribosomal tunnel exit to the SBD β substrate binding cleft, which we tentatively attributed to the nascent chain (Fig. 1d and Supplementary Fig. 1f, red density).~~

9) In the local resolution panels, does red indicate 6.9 Å or does it indicate > 6.9 Å? I mean, for example, when colouring the map, is the range set to a max of 6.9, so any voxels with a value of 6.9 or above are rendered in red? What package was used for the local resolution calculation?

We thank Reviewer #2 for pointing out this mislabeling. Indeed, "red" indicates $\geq 6.9\text{\AA}$ (3x Nyquist limit) and we corrected Supplementary Fig. 3 (now Supplementary Fig. 4) accordingly. CryoSPARC was used for local resolution estimation. We included this information in the Methods section.

~~Particles with more defined tRNA density were selected for homogeneous refinement. The resulting maps were post-processed via DeepEMhancer using the 'high-res' weights provided by the developers⁶³ resulting in cryo-EM maps Ssb-ADP S1 and S2 (Fig. 1b). Local resolution estimation was performed in CryoSPARC⁵⁸.~~

Minor Comments:

1) The company Thermo Fisher Scientific is referred to as both "ThermoFischer Scientific" and "Thermo Scientific" in the methods. It should be consistent and spelt correctly

Thermo Fisher Scientific is now spelled correctly throughout the text.

2) Line 632, the provided magnification is not consistent with Table 2.

We thank the reviewer for pointing out this inconsistency. Magnification was 130,000x. We corrected the magnification in the Methods section.

3) Table 1 indicates the FSC threshold is 0.5, but in the Extended Data Figure 2B and E, it appears that 0.143 is used

The reviewer refers to the “Model resolution,” which was calculated using the phenix validation algorithm. In this procedure, theoretical maps are generated from the structural model and compared to the experimental map at an FSC threshold of 0.5 by default ⁸. However, the global resolution estimate, obtained by gold-standard Fourier shell correlation of the two experimental half-maps, was determined at the conventional threshold of 0.143, as indicated in Supplementary Table 1 and Supplementary Fig. 4b,f.

4) Line 807: In EMPIAR-XYZ, XYZ can be updated.

We updated the EMPIAR code to EMPIAR-12933 in the Data Availability section.

5) Extended Data Fig. 2. The number of particles in the lower part of the State S1 side should be double-checked. The figure indicates that 1.28×10^5 particles go into the 3D classification, then 53% are selected. In the next step after the selection, the particles have increased to 1.31×10^5 and again increase after the next classification to 6.14×10^5 .

We thank the reviewer for pointing out this inconsistency. We now indicate the correct number of particles that went into the initial classification (2.43×10^5) resulting the 1.31×10^5 and 6.14×10^4 particles in the subsequent steps (Supplementary Fig. 2).

6) Extended Data Figure 5 in some binding assays, the fractions are designated T, C, and R and in other T, S, and R (i.e. Extended Data Figure 7). The reason for the difference is unclear.

Reviewer #2 is correct; we made this distinction in the previous version of the manuscript. Our intention was to differentiate T (total), C (cytosol), and R (ribosomal pellet) in experiments analyzing ribosome-binding of Ssb in total cell extracts from T (total), S (supernatant), and R (ribosomal pellet) in experiments analyzing ribosome-binding of Ssb using purified components (i.e., in the absence of cytosolic proteins). However, we recognize that this terminology may be confusing.

To clarify, we now consistently use T (total), S (supernatant), and R (ribosomal pellet) throughout. Whether S refers to cytosolic proteins not associated with ribosomes or to purified proteins not associated with ribosomes is clearly indicated in the Figure Legends, and the graphs specify the y-axes as either “ribosome-bound Ssb in cell extract [%]” or “purified Ssb bound to purified ribosomes [%]”.

7) Extended Data Figure 5D. Is it possible the western blots have been shifted up, as here Ssb and rpl4 run at 116 and 66 kDa, while in Extended Data Figure 5G they run at 66 and 35 kDa

We now included the correct markers for the immunoblots shown in Supplementary Fig. 6d (previous Extended Data Fig. 5D).

8) Extended Data Figure 4 right-hand panels. It would be useful to have the L25 label. We labeled L25 (now Supplementary Fig. 5), thanks for pointing that out.

Reviewer #3 (Remarks to the Author):

Co-translational folding of proteins by chaperones is a critical process in protein homeostasis. Hsp70s are an abundant class of chaperones that interact with and facilitate folding of nascent chains. In yeast, the Hsp70 Ssb has a cochaperone, RAC, comprising Zuo1 and SSz1. How Ssb and RAC function together on the ribosome is not well understood. In this manuscript, the authors use single particle cryo-EM as well as biochemical and genetic approaches to examine this question. The principal finding of the manuscript is a high-resolution structure of Ssb bound to programmed ribosome nascent chain complexes. The authors find that the primary binding site for Ssb1 is the ribosomal protein Rpl25 (uL23). uL23 is at the exit tunnel and this binding site is consistent with previous crosslinking data and by convincing genetic and biochemical experiments presented in this manuscript. The authors attempt to address the question of how Ssb and RAC function is coordinated by using AlphaFold predictions and proposing a complex series of domain rearrangements. However, this is highly speculative and largely leaves open the issue of the coordination between Ssb1 and RAC. Overall, the experiments are cleanly executed and presented, however the manuscript would have greater impact if more concrete experimental evidence were presented for the interplay between Ssb and RAC.

Major points

1. Figure 1 Can the authors comment on what contributes to the differential positioning of SBD- β in the S1 and S2 structures? Does it make specific contacts with the ribosome or are these different conformations intrinsic to Ssb itself?

We thank Reviewer #3 for pointing out that we had insufficiently explained this point in the manuscript. To facilitate better understanding of this important result, we have revised the manuscript accordingly (please see below). Overall, the conformation of the Ssb-SBD in the atomic models Ssb-ADP S1 and S2 does not differ significantly. What differs is the positioning of the Ssb-SBD β relative to the ribosomal surface, with S2 being approximately 19° closer to the surface, as illustrated in Fig. 1f. Additionally, we provide a side-by-side comparison (front and back views) of Ssb-ADP S1 (Supplementary Fig. 5b) and S2 (Supplementary Fig. 5a), detailing the differences in conformation and contacts between the two states.

Changes in the manuscript:

1.

We added the information that the structures of the Ssb-SBD in the S1 and S2 state are similar, however, their positioning on the ribosome differs to the Results Section.

The combined data suggested that both Ssb-ADP S1 and S2 corresponded to ribosome-bound Ssb engaged with a nascent chain substrate (Fig. 1b,d and Supplementary Fig. 1e,f). In both states, Ssb-SBD adopted the closed conformation of an ADP-bound Hsp70 (Supplementary Fig. 1a), but differed in the position of SBD β relative to the translating ribosome, which was 19° closer to the ribosomal surface in S2 when compared to S1 (Fig. 1f and Supplementary Fig. 5a,b).

2.

We improved the Figure Legends of Supplementary Figs. 4a,b (now Supplementary Fig. 5a,b). Which now detail ribosomal contact sites of Ssb more clearly.

Supplementary Fig. 5. Ribosomal contacts of Ssb. (a) Contacts of Ssb with the ribosomal surface in the atomic model Ssb-ADP S2. The RKKR-motif in helix Ssb-αD interacts with the EDD-motif of Rpl25, and Ssb-K567/Ssb-R568 contact the tip of 25S rRNA h59-ES24 (front view). Furthermore, residues within helix Ssb-αA of SBDα and Ssb-SBDβ are close to the ribosomal surface and within crosslinking distance of Rpl35, Rpl19, and Rpl26. Ssb-K530 is 8 Å from Rpl35-K26 and Ssb-K567 is 6 Å from Rpl19-K43 (back view), consistent with *in vitro* NH₂-specific crosslinking⁵. Ssb-R545 forms *in vivo* site-specific crosslinks with Rpl35, Ssb-K497 with Rpl26, and Rpl35-E27/K26/R38 as well as Rpl26-K84/E88 form *in vivo* site-specific crosslinks with Ssb⁶ (back view). Residues 428-430 (KRR-motif) in Ssb-SBDβ contact 5.8S rRNA nucleotides UUC (81-83) (back view), providing a rationale for the moderately reduced ribosome-binding of Ssb mutants in which the KRR motif was replaced with alanine residues⁷. For details, refer to the Results and Supplementary Note 1. (b) Contacts of Ssb with the ribosomal surface in the atomic model of Ssb-ADP S1. Similar to the S2 state shown in a, the RKKR-motif of Ssb-ADP S1 interacts with the EDD-motif of Rpl25, and Ssb-K567/Ssb-R568 are close to h59-ES24 (front view). However, helix Ssb-αA of SBDα and Ssb-SBDβ are positioned farther from the ribosomal surface, with distances of 16 Å between Ssb-K530 and Rpl35-K26, and 11 Å between Ssb-K567 and Rpl19-K43 (back view). For details, refer to the Results and Supplementary Note 1. Color code in a and b: Ssb-SBDβ (forest), Ssb-SBDα, (split pea), Ssb residues involved in ribosome-binding (yellow), Ssb residues involved in site specific crosslinking (brightorange), Rpl25 residues involved in ribosome-binding of Ssb (chocolate), Rpl25 (light magenta), Rpl35 (violetpurple), Rpl26 (marine), Rpl19 (slate blue), Rpl31 (sky blue), Rpl39 (deepblue), helix 59 expansion segment 24 (ES24, pink), 5.8S rRNA (5.8 S, salmon). (c) Amino acid sequence of Ssb-SBDα and mutations within Ssb-SBDα affecting ribosome-binding of Ssb. Upper sequence: Ssb-SBDα consisting of the 4 most C-terminal α-helices, termed Ssb-αA, Ssb-αB, Ssb-αC, and Ssb-αD⁵. Lower sequence: amino acids replaced in Ssb-DD1-DD2 (this work) or Ssb-EE/LBC⁵ are highlighted in green. For details see Results and Supplementary Note 1.

3. We revised Supplementary Note 1 to more clearly highlight the differences and similarities between contacts of Ssb-ADP S1 and Ssb-ADP S2.

Supplementary Note 1. Positioning of Ssb in the atomic models Ssb-ADP S1, Ssb-ADP S2, and in the superimposed Ssb-ATP model. Two major ribosomal contacts of Ssb are shared among the atomic models Ssb-ADP S1, Ssb-ADP S2, and the superimposed model of ribosome-bound Ssb-ATP. First, the electrostatic interactions between helix Ssb- α D (RKKR-motif) and Rpl25 (EDD-motif) in Ssb-ADP S1 (Fig. 2a), Ssb-ADP S2 (Supplementary Fig. 6a), and Ssb-ATP (Supplementary Fig. 7c). These contacts were verified by biochemical analysis (Figs. 2,3). Second, the contact formed between Ssb-K567/Ssb-R568 in the loop connecting Ssb- α B and Ssb- α C (Supplementary Fig. 5c) and 25S rRNA h59-ES24 (Supplementary Fig. 5a,b and Supplementary Fig. 7c). The latter contact is consistent with previous *in vivo* crosslinking and analysis of cDNA (CRAC) data⁵, and with the observation that ribosome-binding of Ssb carrying the K567E/R568E mutations is reduced by more than 50%.⁵. In the Ssb-ADP S2 atomic model, helix Ssb- α A of SBD α (Supplementary Fig. 5c) and the entire Ssb-SBD β are positioned closer to the ribosomal surface than in Ssb-ADP S1 and the Ssb-ATP models. Specifically, Ssb-ADP S2 is in close proximity to Rpl35, Rpl19, and Rpl26, whereas Ssb-ADP S1 is not (Supplementary Figs. 5a,b). This suggests that (i) *in vitro* crosslinking obtained with the amino-specific crosslinker BS³ (spacer length of ~11 Å) between Ssb and Rpl35/Rpl19⁵; and (ii) *in vivo* photo-crosslinking (spacer length of ~12 Å,¹³) of Ssb-R545 to Rpl35, Ssb-K497 to Rpl26, and Rpl26-K84, Rpl26-E88, Rpl35-E27, Rpl35-R38 to Ssb⁶ originate from ribosomes with Ssb-ADP bound in the S2 state, or in a similar conformation (Supplementary Figs. 5a). Moreover, residues 428-430 (KRR-motif) in Ssb-SBD β are close to 5.8S rRNA nucleotides UUC (81-83) in the atomic model of Ssb-ADP S2, but not in S1 (Supplementary Fig. 5a,b back view). Previous mutational analysis revealed that replacement of the KRR-motif with alanine residues moderately reduced ribosome-binding of Ssb⁷, suggesting that Ssb-ADP S2 binds the ribosome more stably than Ssb-ADP S1.¶

2. In Figure 3f (now Fig. 3g), the salient question is whether or not the residual ribosome binding activity of Ssb-DD1-DD2 is due to Ssz1. The synthetic growth defect is suggestive but indirect. The authors should directly test if Ssz1 is responsible for the residual Ssb binding by measuring binding of Ssb1-3*-DD1-DD2 to ribosomes or the binding of Ssb-DD1-DD2 to ribosomes in the absence of Ssz1. To support the genetic interaction, the authors should also show that this protein expresses well, so that the binding and growth defects cannot be attributed to poor expression of the mutant protein.

We thank Reviewer #3 for these excellent suggestions.

We now show the expression level of Ssb-3*-DD1-DD2 in the new Supplementary Fig. 7d. The expression of Ssb-3*-DD1-DD2 resembles the expression level of wild type Ssb.

We now directly compared ribosome-binding of Ssb-DD1-DD2 and Δ ssz1 Ssb-DD1-DD2 side by side. We also included the Ssb1-3* DD1-DD2 mutant in the analysis: in this mutant binding of Ssb to both Rpl25 and the Ssz1-NBD is impaired. The results are shown in the new Fig. 3h and new Supplementary Fig. 7e.

In summary, residual ribosome binding of Ssb-DD1-DD2 (Fig. 3b,c) is further reduced when Ssb carries mutations that affect its interaction with Ssz1 or when Ssz1 is absent. We have included this important finding in the Results section.

Results:

Direct ribosome-binding of Ssb-DD1-DD2 was severely impaired (Fig. 3b,c), yet did not affect yeast growth (Supplementary Fig. 7a). It was previously shown that Ssb-ATP can also associate with ribosomes indirectly, through interaction of the Ssb-NBD with the NBD of the RAC subunit Ssz1²⁸. Mutations in Ssb (R261D/E370R/E547R, termed Ssb-3*), which significantly reduce indirect ribosome association of Ssb via Ssz1, likewise did not affect yeast growth under normal conditions²⁸. We therefore asked whether a combination of the Ssb-DD1-DD2 with Ssb-3* mutations might result in synthetic growth defects. The Ssb-3* mutations caused moderate growth defects in our strain background (Fig. 3g and Supplementary Fig. 7d). Importantly, growth defects were significantly enhanced in a strain expressing Ssb-3*-DD1-DD2, which combines the two sets of mutations (Fig. 3g and Supplementary Fig. 7d). The synthetic growth defect suggested that proper function of the RAC-Ssb cycle was dependent on positioning of Ssb-ATP at the ribosome, either through the interaction of Ssb with Rpl25 or through interaction between the NBDs of Ssb and Ssz1. To test this possibility, ribosome-binding of Ssb-DD1-DD2 was compared with that of the Ssb-3*-DD1-DD2 mutant and with Ssb-DD1-DD2 expressed in the Δ ssz1 background (Fig. 3h and Supplementary Fig. 7e). According to the dual binding site model, the latter two mutants should exhibit markedly reduced binding. Indeed, ribosome-binding of Ssb-3*-DD1-DD2 and of Ssb-DD1-DD2 expressed in the Δ ssz1 background was further reduced by more than fivefold (Fig. 3h and Supplementary Fig. 7e) compared to the already strongly reduced ribosome-binding of Ssb-DD1-DD2 (Fig. 3b,c). These findings demonstrate that residual ribosome-binding observed for Ssb-DD1-DD2 is mediated by the interaction between the Ssb-NBD and Ssz1.¶

3a. In Figure 5, the authors explain that Ssb and Rac initially bind simultaneously, followed by a hand off to Ssb. They also argue that Ssz1 provides a binding site for Ssb1, see figure 3f (now 3g). However, the data in figure 4d argue that the two complexes are not present on the ribosome at the same time. These results seem inconsistent.

These observations, which may initially appear contradictory, arise from markedly distinct conformations that Ssb adopts when bound to ATP (PDB 5TKY⁵) versus when bound to ADP (PDB 2KHO⁴, and Fig. 1b of this work). The transition between these conformational states, triggered by ATP hydrolysis, occurs rapidly. We have now included this essential background information more clearly in the Introduction (see below).

The different conformations of Ssb are depicted side-by-side in Supplementary Figs. 9a (ribosome-bound Ssb-ATP), 9c (ribosome-bound Ssb-ADP S1), and 9e (ribosome-bound Ssb-ADP S2). RAC can be accommodated on ribosome-Ssb-ATP complexes (Supplementary Fig. 9b), and possibly also on Ssb-ADP S1 complexes; however, Ssb-ADP S1 and RAC are already in close proximity (Supplementary Fig. 9d). RAC, however, cannot be accommodated on Ssb-ADP S2 complexes due to steric clashes between the Ssz1-NBD and Ssb-SBD β (Supplementary Fig. 9f).

Of note, our fluorescence anisotropy measurements revealed nearly identical K_d values for Ssb binding to ribosomes and ribosome-RAC complexes when Ssb is in the open conformation resembling the ATP-bound state (Supplementary Fig. 8f). Moreover, phenotypic analysis of the Ssb-DD1-DD2 mutant, the Ssb-3*-DD1-DD2 mutant, and the Ssb-DD1-DD2 mutant in the Δ ssz1 background (Fig. 3g), as well as analysis of ribosome binding by these mutants (new Fig. 3h), are consistent with conformational states of the RAC-Ssb system in which RAC and Ssb can be accommodated on the same ribosome. However, after ATP hydrolysis, when Ssb is bound to ADP, Ssb and RAC can no longer bind concurrently to the same ribosome. This is demonstrated by the experiment shown

in Fig. 4d and is consistent with the structural models provided in Supplementary Fig. 9.

Based on the combined data, we propose the model depicted in Fig. 5 in that RAC release occurs upon ATP hydrolysis by Ssb, because, in its new position, Ssb-SBD β interferes with RAC binding (Supplementary Fig. 9f). According to this model, RAC release occurs exactly when RAC has fulfilled its dual function: (i) positioning Ssb-SBD β close to the emerging nascent chain and (ii) catalyzing ATP hydrolysis, which leads to stable binding of Ssb to the nascent chain - and RAC's own release.

3b. The authors generated a nice set of truncated PGK1 vectors that allows the production of RNCs of different lengths. Can they show that RAC does bind on short nascent chains and that Ssb predominates as chains grow in length?

We are pleased to refer Reviewer #3 to our previous study published in Nature Comm. in 2020 ¹ under the title: "*The ribosome-associated complex RAC serves in a relay that directs nascent chains to Ssb*". In Fig. 1 of Zhang et al. 2020 we show that Zuo1 and Ssz1 contact nascent chains prior to Ssb. For your convenience we included Fig. 1c and the experimental set up, which is detailed in Fig. 1b (Zhang et al. 2020). Fig. 1c and additional results of Zhang et al. 2020 reveal that the emerging nascent chain replaces the Zuo1 LP-motif from the peptide binding cleft of Ssz1-SBD β and provide functional insight into the coordination between Ssb and RAC on translating ribosomes.

Figure 1c Zhang et al. 2020 ¹ [redacted]

(b) Experimental set up of crosslinking experiments. Isolated RNCs carrying [³⁵S]-labelled nascent chains are crosslinked to adjacent proteins using the homobifunctional amino-reactive crosslinker BS³, spacer length 11.4 Å. Crosslink products between nascent chains and Ssb, Ssz1, or Zuo1 are then identified via immunoprecipitation under denaturing conditions using antibodies directed against Ssb, Ssz1, or Zuo1.

(c) Contacts of Ssb, Zuo1, and Ssz1 with short nascent chains. RNCs carrying P_{gk1-40} (40 residues), -45 (45 residues), or -50 (50 residues) were generated in a wild type translation extract.

We now present these previous findings more clearly in the Introduction, as they provide the basis for the model proposed in Fig. 5.

rudimentary Ssz1-SBD β .^{6,18-20} However, when bound to ribosomes, RAC adopts a highly flexible structural arrangement, as evidenced by the markedly different conformations observed in ribosome-bound RAC through cryo-EM studies.^{16,17} Conformational flexibility within RAC is highlighted by the extreme N-terminus of Zuo1, termed the LP-motif (Zuo1-LP; Supplementary Fig. 1b²¹), which binds to Ssz1-SBD β as a pseudo-substrate but is displaced upon nascent-chain binding.²¹ Displacement of the Zuo1-LP occurs as soon as the nascent chain emerges from the ribosomal tunnel, thereby loosening the Zuo1-Ssz1 interaction and permitting conformational rearrangements of the N-terminal Zuo1 domain (Zuo1-ND; Supplementary Fig. 1b). Shortly thereafter, the nascent chain is handed over from Ssz1-SBD β to Ssb-SBD β , but how this transfer is coordinated remains unknown.²¹ ¶

Minor points

4. Panel D of Figure 2 is conceptually unrelated to the rest of the figure and should be moved to supplementary data. Furthermore, the idea being tested here is that mutations in Rpl25 affect SRP binding. However, the readout is quite indirect - whether or not expression of a substrate for SRP is affected. Directly examining SRP binding to Rpl25-KKK ribosomes would be more direct.

We agree with Reviewer #3 that SRP binding represents a side result in the context of this study. As requested, we moved Fig. 2d to the Supplementary Figures, where it is now presented as Supplementary Fig. 6i. However, we consider it important to provide an explanation for the strong growth defect of the Rpl25-KKK strain (Supplementary Fig. 6b). This severe growth defect is consistent with a role of Rpl25 in SRP-dependent targeting to the ER and is supported by the observation that Dap2, a membrane protein which requires SRP for targeting, is not properly delivered in the Rpl25-KKK strain (Supplementary Fig. 6h).

With regard to directly examining SRP binding to Rpl25-KKK ribosomes, we appreciate the reviewer's suggestion. Unfortunately, with the methods currently available analysis of SRP binding to Rpl25-KKK ribosomes was not feasible in the context of this study for the following reasons.

The bulk of SRP is lost during the preparation of total extract, which is used for ribosome-binding experiments (Fig. R2). Of the residual SRP in the total extract, only a minor fraction is associated with ribosomes⁹, likely because SRP is not stably associated with translating ribosomes during ribosome scanning^{10,11}. Thus, ribosome-binding experiments, as those performed with Ssb and RAC in this study, are not suited to compare the binding of SRP to wild type and Rpl25-KKK ribosomes. A valid comparison could be performed using *in vitro* translation followed by crosslinking and immuno-purification of the crosslinking products. We have previously performed this type of analysis with wild type translation extract and found that SRP is efficiently recruited to RNCs attached to Dap2, but not Pgk1¹¹. However, the approach requires a translation extract prepared from the Rpl25-KKK strain, which we were unable to generate due to the severe slow-growth phenotype of this mutant strain (Supplementary Fig. 6b).

Figure R2. Total extract (soluble material prepared from yeast cells after glass beads disruption) and cell lysate (yeast cells lysed under denaturing conditions) was prepared as described in the manuscript. Loading was adjusted to equalize ribosome loading (Rpl4).

To acknowledge the limitations of our SRP data, we have carefully revised and moderated our statement in the Results section and more clearly indicated the purpose of the SRP-related experiments. Please note that we do not claim that SRP does not bind to Rpl25-KKK ribosomes, but rather focus on the effect that could underlie slow growth of the Rpl25-KKK strain.

(Supplementary Fig. 6j³⁷). This pattern changed in the absence of SRP: under this condition Ssb was efficiently crosslinked to nascent Dap2 (Supplementary Fig. 6i). The data support a model in which SRP and Ssb compete for an overlapping ribosomal attachment site involving the Rpl25-EDD motif. They further suggest that the competition of SRP and Ssb for nascent chains, observed in a global approach³⁸, is unlikely to result from a general inability of Ssb to interact with SRP substrates, but may rather reflect direct competition at the tunnel exit. Importantly, the impaired delivery of the SRP-dependent ER model protein Dap2 (Supplementary Fig. 6h) provides a plausible explanation for the severe growth defects observed in strains carrying mutations in the Rpl25-EDD motif (Supplementary Fig. 6b).¶

5. Figure 3 introduces the DD1-DD2 mutant. Including a cartoon of this region of Ssb and the substitutions in the figure would help the reader understand exactly what the mutation is without having to refer to the supplemental materials.

This is an excellent suggestion. We now show a cartoon of helix SBD- α D as Fig. 3a.

6. Figure 4a Because of the scale of the x-axis, the curves are not well separated. Including an inset for ribosome concentration range $<1\mu\text{M}$ would be helpful.

Reviewer #3 is correct. We initially tried using an inset, but it provided little improvement. We now display the x-axis on a logarithmic scale as typical in the field. The inflection point of the sigmoidal curves on the log-scaled axis allows an approximate visual estimation of the K_d .

7. L44 Please explain the composition of RAC here rather than referring the reader to the supplemental materials.

We have extended the Introduction to provide a more detailed description of the structure of RAC.

Ssb is critically important for cotranslational protein folding, as it binds directly to ribosomes via SBD α ^{5,7} and, from this position, accesses newly synthesized polypeptides⁸⁻¹⁰. To perform its function Ssb, requires a unique cochaperone, termed ribosome-associated complex (RAC)^{6,11,12}. The heterodimeric RAC consists of the JD-protein Zuo1¹³ and the Hsp70 homolog Ssz1^{6,11}. The Zuo1 subunit interacts with the 60S as well as 40S ribosomal subunit^{6,14-17} and possesses a unique domain structure^{6,16,17} (Supplementary Fig. 1b). Ssz1 is a non-canonical Hsp70 as it binds, but does not hydrolyze ATP, and lacks the C-terminal portion of SBD β and the complete SBD α ^{6,18,19} (Supplementary Fig. 1b). In the absence of ribosomes, Ssz1 and Zuo1 engage in a tight interaction, mediated by the N-terminal domain of Zuo1 and the rudimentary Ssz1-SBD β ^{6,18-20}. However, when bound to ribosomes, RAC adopts a highly flexible structural arrangement, as evidenced by the markedly different conformations observed in ribosome-bound RAC through cryo-EM studies^{16,17}. Conformational flexibility within RAC is highlighted by the extreme N-terminus of Zuo1, termed the LP-motif (Zuo1-LP; Supplementary Fig. 1b²¹), which binds to Ssz1-SBD β as a pseudo-substrate but is displaced upon nascent chain binding²¹. Displacement of the Zuo1-LP occurs as soon as the nascent chain emerges from the ribosomal tunnel, thereby loosening the Zuo1-Ssz1 interaction and permitting conformational rearrangements of the N-terminal Zuo1 domain (Zuo1-ND; Supplementary Fig. 1b). Shortly thereafter, the nascent chain is handed over from Ssz1-SBD β to Ssb-SBD β , but how this transfer is coordinated remains unknown²¹. ¶

8. Please explain how ADP was assigned in the cryo-EM structures if the nucleotide binding domain was not resolved.

We apologize for having omitted this essential information and thank the Reviewers for drawing our attention to this important point. As it was raised by all three Reviewers, our response is addressed to Reviewers #1-3.

Reviewer 1: 3. How was ADP assigned to the ribosome-bound Ssb in the structure? Was this by inference or by structural modeling? This reasoning could be clearer.

Reviewer 2: 3) The text indicates that Ssb is in the ADP form, but it is unclear if ADP or ATP is in the buffer or co-purifies with the protein. I guess the ADP form is inferred from the domain arrangement of Ssb, but this could be clarified in the text.

Reviewer 3: 8. Please explain how ADP was assigned in the cryo-EM structures if the nucleotide binding domain was not resolved.

The nucleotide binding domain (NBD) of Ssb was not resolved in the Ssb-ADP S1/S2 structures; therefore, the nucleotide bound to the Ssb-NBD could not be assessed directly. Instead, we inferred the nucleotide state from the conformation of the substrate-binding domain (SBD) in the cryo-EM maps (Fig. 1b). In both Ssb-ADP structures, the conformation of the Ssb-SBD closely resembled that of the ADP-bound DnaK-SBD (PDB 2KHO⁴). Specifically, Ssb-SBD α (the “lid domain”) was observed in a closed position over the Ssb-SBD β substrate-binding cleft. By contrast, the SBD of ATP-bound Ssb (and Hsp70s in general) adopts a distinct open conformation (PDB 5TKY⁵), which was incompatible with the conformation of ribosome-bound Ssb observed in our cryo-EM maps.

We have improved the Introduction by providing a more detailed description of the distinct conformations of Hsp70 proteins and their functional significance. In the Results section, we now explicitly state that the conformation of the Ssb-SBD is consistent exclusively with the ADP-bound state.

Introduction:

(Supplementary Fig. 1a). Yeast Ssb is a canonical Hsp70 (encoded by the nearly identical *SSB1* and *SSB2* genes), consisting of an N-terminal nucleotide-binding domain (NBD) and a C-terminal substrate-binding domain (SBD) (Fig. 1a). The NBD possesses ATPase activity; the SBD is subdivided into a β -sheet domain (SBD β) and an α -helical lid domain (SBD α) (Fig. 1a). As in other canonical Hsp70s, the two domains of Ssb are connected via a linker that allows allosteric coupling of ATP hydrolysis with tight substrate binding to the SBD (Supplementary Fig. 1a). When Ssb is bound to ATP, the linker interacts with the NBD, and the SBD α lid adopts an open conformation⁵. Upon ATP hydrolysis, significant structural rearrangements occur, which lead to the detachment of the linker from the NBD, allowing the SBD α lid to close over the substrate-binding pocket on SBD β , thereby stabilizing substrate binding.^{1,4,6}¶

Results:

Cryo-EM structure of ADP-bound Ssb associated with translating ribosomes. To determine the structure of Ssb associated with translating ribosomes, ribosome-nascent chain complexes (RNCs) carrying FLAG-tagged 3-phosphoglycerate kinase (FLAG-Pgk1-70; Supplementary Fig. 1c-f) were generated in a yeast *in vitro* translation system in the presence of Ssb, were subsequently purified by FLAG affinity chromatography, and were analyzed by cryo-EM (Supplementary Fig. 2, see Methods). In these samples, ribosomes representing diverse states along the translational cycle were detected. Notably, non-rotated RNCs containing a P-site tRNA and associated with the density of a trailing ribosome within a polysome displayed enhanced density definition near the ribosomal tunnel exit (Supplementary Fig. S3). Further analysis of these particles (Supplementary Fig. S2) yielded two cryo-EM structures in which Ssb was bound to ribosomes stalled in a non-rotated, post-translocation conformation with a P-site Met-elongator tRNA (tRNA^{Met-e}). In both structures, the SBD of Ssb (Fig. 1b and Supplementary Table 1) was in the closed conformation characteristic for the ADP-bound state of Hsp70s, in which SBD α is tightly packed against SBD β (see Introduction). The Ssb-NBD was not visible in the cryo-EM maps, consistent with the known flexibility between the NBD and SBD in ADP-bound Hsp70s.^{4,29,30} The resolution of the complexes,

9. In figure 5, the cartoons are too small to convey the conformational changes being proposed.

We have enhanced the size of the cartoons in the model shown in Fig. 5 and also simplified the model significantly. Detailed views of ribosome-bound RAC and Ssb in the different conformational states are also displayed in Supplementary Figs. 9 and 10.

General information for Reviewers #1-3

During the revision process, we corrected the molecular models of Ssb-ADP S1 and S2. The C-terminal carboxyl group of Rpl25 had been mistakenly modeled as a hydroxyl group; this error has been corrected. The Rpl25 C-terminal carboxyl group is important for the interaction between Rpl25 and Ssb (Fig. 2a and Supplementary Fig. 6a). Furthermore, in the initial ribosome model (PDB 6T7I), incorrect sequences were linked to Rpl6b and Rpl24b (Fig. R3). The Rpl6b and Rpl24b sequences have been corrected. The revised models have been re-uploaded to the PDB, and Supplementary Table 1 has been updated accordingly.

a

PDB-6T7I Rpl6b sp P05739 RL6B_YEAST	MTAQQAPKWYPSEDVAAPKKTRKAVRPQKLRSASLVPGTVLILLAGRFRGKRVVYLKHLED MTAQQAPKWYPSEDVAAPKKTRKAVRPQKLRSASLVPGTVLILLAGRFRGKRVVYLKHLED *****
PDB-6T7I Rpl6b sp P05739 RL6B_YEAST	NTLLVTGPFKVNGLPLRRVNARYVIATSTKVSVEGVNVEKFNVEYFAKEKLTKEKKEAN NTLLVTGPFKVNGLPLRRVNARYVIATSTKVSVEGVNVEKFNVEYFAKEKLTKEKKEAN *****
PDB-6T7I Rpl6b sp P05739 RL6B_YEAST	LFPEQQTKEIKTERVEDQKVVDKALLAEIKKTPLLKQYLSASFSLKNGDKPHMLKF LFPEQQTKEIKTERVEDQKVVDKALLAEIKKTPLLKQYLSASFSLKNGDKPHLLKF *****:*****:***

b

PDB-6T7I Rpl24b sp P24000 RL24B_YEAST	MKVEIDSFSGAKIYPGRGTLFVRGDSKIFRFQNSKSASLFKQRKNPRRIA MKVEVDSFSGAKIYPGRGTLFVRGDSKIFRFQNSKSASLFKQRKNPRRIA *****
PDB-6T7I Rpl24b sp P24000 RL24B_YEAST	WTVLFRKHKKGITTEEVAKKRSRKTVKAQRPITGASLDLIKERRSLKPEV WTVLFRKHKKGITTEEVAKKRSRKTVKAQRPITGASLDLIKERRSLKPEV *****
PDB-6T7I Rpl24b sp P24000 RL24B_YEAST	RKAQREEKQKADKEKKKAKAARKAEKAKSAGVQGSKVSQKQAKGAFQKV RKANREEKLNKKEKRAEKAARKAEKAKSAGVQGSKVSQKQAKGAFQKV *****:*****:*****
PDB-6T7I Rpl24b sp P24000 RL24B_YEAST	AATSR AATSR *****

Figure R3. Clustal ω alignment of (a) Rpl6b and (b) Rpl24b sequences in PDB 6T7I and the corresponding sequence from UniProt.

References

1. Zhang, Y. *et al.* The ribosome-associated complex RAC serves in a relay that directs nascent chains to Ssb. *Nat. Commun.* **11**, 1504 (2020).
2. Chen, Y., Tsai, B., Li, N. & Gao, N. Structural remodeling of ribosome associated Hsp40-Hsp70 chaperones during co-translational folding. *Nat. Commun.* **13**, 3410 (2022).
3. Sanchez-Garcia, R. *et al.* DeepEMhancer: a deep learning solution for cryo-EM volume post-processing. *Commun Biol* **4**, 874 (2021).
4. Bertelsen, E. B., Chang, L., Gestwicki, J. E. & Zuiderweg, E. R. Solution conformation of wild-type E. coli Hsp70 (DnaK) chaperone complexed with ADP and substrate. *Proc. Natl. Acad. Sci. U S A* **106**, 8471-8476 (2009).
5. Gumiero, A. *et al.* Interaction of the cotranslational Hsp70 Ssb with ribosomal proteins and rRNA depends on its lid domain. *Nat. Commun.* **7**, 1-12 (2016).
6. Verghese, J. & Morano, K. A. A lysine-rich region within fungal BAG domain-containing proteins mediates a novel association with ribosomes. *Eukaryot. Cell* **11**, 1003-1011 (2012).
7. Tesina, P. *et al.* Molecular mechanism of translational stalling by inhibitory codon combinations and poly(A) tracts. *EMBO J.* **39**, e103365 (2019).
8. Afonine, P. V. *et al.* New tools for the analysis and validation of cryo-EM maps and atomic models. *Acta Crystallogr D Struct Biol* **74**, 814-840 (2018).
9. Raue, U., Oellerer, S. & Rospert, S. Association of protein biogenesis factors at the yeast ribosomal tunnel exit is affected by the translational status and nascent polypeptide sequence. *J. Biol. Chem.* **282**, 7809-7816 (2007).
10. Flanagan, J. J. *et al.* Signal recognition particle binds to ribosome-bound signal sequences with fluorescence-detected subnanomolar affinity that does not diminish as the nascent chain lengthens. *J. Biol. Chem.* **278**, 18628-18637 (2003).
11. Berndt, U., Oellerer, S., Zhang, Y., Johnson, A. E. & Rospert, S. A signal-anchor sequence stimulates signal recognition particle binding to ribosomes from inside the exit tunnel. *Proc. Natl. Acad. Sci. U S A* **106**, 1398-1403 (2009).

Reply to Reviewers

Reviewer #1 (Remarks to the Author):

The revised manuscript addresses my comments and I support publication.

We thank Reviewer #1 for his/her positive evaluation.

Reviewer #2 (Remarks to the Author):

I would like to thank the authors for their thorough response and revision. The manuscript has improved in both clarity and rigour. From my side, the responses to the review comments are detailed and convincing, and I find that my previous concerns have been largely addressed. I have no further major comments.

We thank Reviewer #2 for his/her positive response. We also feel that the manuscript is improved thanks to the input of the Reviewers.

A minor question/clarification would be how the rotation axis in Supplementary Fig. 3D was determined. Also in Figure 3 are the volumes shown from a backprojection (cryodrgn backproject_voxel) of all particles in the class, or the volume at the centre of the class?

1. Determination of the rotation axis

The axes connect the tip of expansion segment 6 (ES6) of the 18S rRNA to the C-terminus of the 40S ribosomal protein Asc1. The black axis represents the orientation in the non-rotated state, and the gray axis shows the rotation of the 40S subunit relative to this state, with the 60S subunit held fixed.

We now specify how the rotated axis was determined in the Legend of Supplementary Fig. 3d.

(d) Rotational state of the 40S ribosomal subunit in clusters 1 to 8. The axes connect the tip of expansion segment 6 (ES6) of the 18S rRNA to the C-terminus of the 40S ribosomal protein Asc1. Non-rotated axis (black line), rotated axis (gray line), with the 60S subunit held fixed.

2. cryoDRGN analysis

In the cryoDRGN analysis, we performed kmeans clustering on the particles latent space embeddings. For this purpose, 1000 volumes were generated via back projection of all particles within a cluster. Subsequently, these 1000 reconstructed volumes were clustered using the agglomerative clustering approach of the cryoDRGN landscape analysis tool. The volumes shown in Supplementary Fig. 3 are centroids, representing a voxel-based average of all reconstructions that were grouped by agglomerative clustering.

We now explain the volume generation in more detail in the Methods section.

Using the cryoDRGN landscape' tool, latent space encodings were first clustered via kmeans (k = 1000) and the resulting reconstructions, generated by back projection, were further clustered via agglomerative clustering (k = 20) (<https://ez-lab.gitbook.io/cryodrgn/>). The

volumes of the resulting clusters are shown as centroids, representing voxel-based averages of the reconstructions within each cluster (Supplementary Fig. 3a,b).